# Environmental fluctuations accelerate molecular evolution of thermal tolerance in a marine diatom

C.-Elisa Schaum[1,2], A. Buckling[1], N. Smirnoff [3], D.J. Studholme [3] & G. Yvon-Durocher[1]

Diatoms contribute roughly 20% of global primary production, but the factors determining their ability to adapt to global warming are unknown. Here we quantify the capacity for adaptation to warming in the marine diatom *Thalassiosira pseudonana*. We find that evolutionary rescue under severe (32 °C) warming is slow, but adaptation to more realistic scenarios where temperature increases are moderate (26 °C) or fluctuate between benign and severe conditions is rapid and linked to phenotypic changes in metabolic traits and elemental composition. Whole-genome re-sequencing identifies genetic divergence among populations selected in the different warming regimes and between the evolved and ancestral lineages. Consistent with the phenotypic changes, the most rapidly evolving genes are associated with transcriptional regulation, cellular responses to oxidative stress and redox homeostasis. These results demonstrate that the evolution of thermal tolerance in marine diatoms can be rapid, particularly in fluctuating environments, and is underpinned by major genomic and phenotypic change.

---

[1] Environment and Sustainability Institute, University of Exeter, Penryn Campus, Penryn, Cornwall TR10 9EZ, UK. [2] Institute for Hydrobiology and Fisheries, Section Oceanography, Hamburg University, 22767 Hamburg, Germany. [3] Biosciences, College of Life and Environmental Sciences, Geoffrey Pope Building University of Exeter, Exeter EX4 4QD, UK. Correspondence and requests for materials should be addressed to C.-E.S. (email: elisa.schaum@uni-hamburg.de) or to G.Y-D. (email: g.yvon-durocher@exeter.ac.uk)

Earth system models predict that global warming will result in significant declines in net primary production by marine phytoplankton throughout the 21st century (up to 20%)[1,2] driven by rising temperatures exceeding limits of thermal tolerance and increases in grazing and nutrient limitation in warmer, more stratified oceans[3–5]. Current models, however, do not consider the potential for marine phytoplankton to rapidly adapt to environmental changes associated with global warming[6–8]. Such shortcomings have unknown consequences for projected changes in global ocean primary production and arise because the mechanisms that facilitate or constrain the capacity for rapid adaptation to warming in marine phytoplankton are largely unknown.

Here, we address this fundamental knowledge gap by carrying out a 300-generation selection experiment, with the model marine diatom *Thalassiosira pseudonana*, to assess the potential for, and mechanisms that might facilitate rapid adaptation to warming in this globally important phytoplankton[9–11]. A key hypothesis we test is that rapidly fluctuating temperatures—an intrinsic feature of natural environments—will play a key role in adaptation[12–14]. Temporary exposure to a benign environment resulting from temperature fluctuations could both accelerate adaptation to severe conditions by increasing the population size (a positive demographic effect) or constrain adaptation, by relaxing selection for beneficial mutations that promote persistence in the harsh environment (a negative population genetic effect)[12,13]. Our experiment was initiated with a single clone (the ancestor), which had an upper thermal limit of 35 °C. The ancestor was then distributed among four experimental treatments that represent a range of warming scenarios (Supplementary Figure 1 for experimental set-up): (i) a control at 22 °C, which was the long-term culture temperature; (ii) moderate warming at 26 °C; (iii) severe warming at 32 °C, and (iv) a fluctuating thermal regime, which cycled between 22 and 32 °C every 3–4 generations. The design of the fluctuating treatment did not set out to mimic temperature variation in the natural environment, but rather serves as a means to contrast evolutionary responses to warming under stable versus fluctuating conditions, where temperatures vary predictably within a timescale that acclimation can adjust the phenotype to cope with the change in environment[15]. To understand the mechanisms that set the limits of thermal tolerance in marine diatoms and quantify the capacity for adaptation beyond present limits, we re-sequenced the genomes, measured growth rate, photosynthesis, respiration, and elemental composition in the ancestor and evolved lineages after approximately 300 generations of selection. We found that adaption to warming was linked to major phenotypic changes in metabolic traits and elemental composition and consistent with these phenotypic changes, the most rapidly evolving genes were associated with transcriptional regulation, cellular responses to oxidative stress, and redox homeostasis.

## Results

### Evolution of thermal tolerance.
Trajectories of population growth rate ($\mu$, day$^{-1}$) over the course of the selection experiment differed substantially between the selection regimes (Fig. 1; Supplementary Tables 1–3). In the control lineages, growth rates increased gradually, presumably reflecting continual laboratory adaptation. However, trajectories of growth rate in the warming treatments were markedly different from the control and one another. Lineages selected under severe warming (32 °C) exhibited a characteristic pattern of evolutionary rescue (i.e., where evolution can reverse population decline owing to environmental stress, and prevent otherwise inevitable extinction)[14,16–18]. After an initial increase in the first 3 weeks of the experiment, growth

rates (and population densities, see Supplementary Figure 2) declined in the 32 °C environment and remained very low (0.24 ± 0.09 s.e.m.) for more than 1 year (~ 100 generations). After approximately 100 generations, growth rates rapidly increased and were statistically indistinguishable from those in the control environment after 300 generations (0.63 ± 0.05 s.e.m. at 32 °C, 0.77 ± 0.11 s.e.m. at 22 °C). Under moderate warming (26 °C), and in the regime that fluctuated between 22 and 32 °C growth rate showed an immediate and sustained increase (2.1 and 1.9-fold faster than the ancestor respectively). These results yield two important insights that are pertinent for understanding the evolutionary dynamics of *T. pseudonana* in response to warming. First, adaptation to severe warming was slow, with evolutionary rescue taking over a year to restore growth rates to levels comparable to the ancestor in the control environment. Second, in the fluctuating environment where populations experienced short bursts of exposure to 32 °C followed by periods in the benign (22 °C) environment, adaptation to the severe environment was rapid. Consistent with our hypothesis, lineages selected under the fluctuating regime maintained substantially larger population sizes relative to those experiencing severe warming (Supplementary Figure 2 and Supplementary Tables 1 and 3), suggesting that temporary restoration of benign conditions increased the probability of fixing beneficial mutations required for adaptation to the severe (32 °C) environment via a positive demographic effect.

To assess whether adaptation to the various selection regimes changed the thermal tolerance curves in the evolved lineages, we quantified growth rates across a temperature gradient spanning 15–40 °C. Populations selected under the moderate (26 °C), severe (32 °C), and fluctuating (22 °C/32 °C) warming treatments had higher optimal growth temperatures, $T_{opt}$, compared to the ancestor and the control (Fig. 1b; Supplementary Tables 1, 4 and 5). Despite having a high $T_{opt}$, the lineages selected under constant severe warming had the slowest growth rates at all measurement temperatures except for 40 °C, the most extreme temperature, suggesting that these populations were still under high-temperature stress in spite of the apparent evolutionary rescue event and the passage of 300 generations of selection. In the fluctuating treatments, growth rate at high temperatures was traded-off against slow growth at low temperature, suggesting that the severe environment was the dominant driver of selection. By contrast, high temperature tolerance in the moderate warming treatments appeared to incur no cost in performance at low temperature. A key question therefore is: what mechanisms facilitated the rapid evolution of increased thermal tolerance in the lineages selected under moderate and fluctuating warming?

### Metabolic efficiency facilitates evolution of thermal tolerance.
The fraction of photosynthetically derived carbon that can be allocated to growth tends to decline with warming, owing to the high temperature sensitivity of respiration relative to photosynthesis, suggesting that the upper thermal tolerances of phytoplankton reflect metabolic constraints that limit the efficiency of allocating metabolic energy to growth at high temperature[19]. To investigate whether changes in metabolic traits could help explain the thermal tolerance curve of the ancestor and evolutionary shifts observed over the selection experiment, we measured responses of gross photosynthesis (P) and respiration (R) to acute gradients in temperature, spanning 4–45 °C, in the ancestor and all evolved lineages. In contrast to previous work[19], the activation energies characterizing the temperature sensitivities of P and R in the ancestor were equivalent (e.g., $E_a$ for P = 1.08 eV ± 0.18 s.e.m., and $E_a$ for R = 1.07 ± 0.17 s.e.m.), i.e., increases in P and R up to the optima were similar (Fig. 2; Supplementary Tables 6–8). Optimum temperatures for P were,

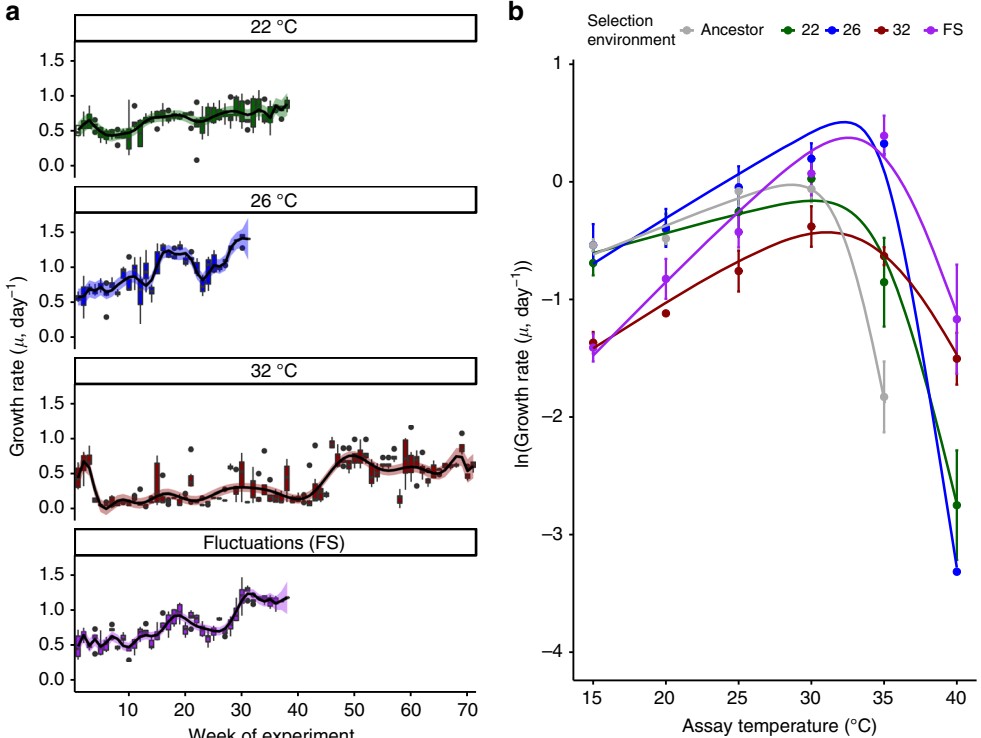

**Fig. 1** Growth rate trajectories and rapid evolutionary shifts in thermal tolerance curves. **a** Time-series of population growth rates reveal immediate and sustained increases in growth rates in populations adapting to moderate warming and the environment that fluctuated between 22 (benign conditions) and 32 °C (severe conditions). In the 32 °C environment populations performed very poorly for approximately 1 year (~100 generation), followed by an evolutionary rescue event, which caused growth rates to increase to levels comparable with the control treatment at 22 °C. Boxplots were created by binning growth rate estimates across replicates on each week of the experiment until 300 generations had passed in each lineage. Fitted lines are from the best fits of a GAMM (see Methods and Supplementary Table 2), with shaded areas indicating 1 s.e.m. and boxplots are displayed so that whiskers indicate 1.5 × interquartile range. **b** Thermal response curves of the ancestor and evolved lineages show marked shifts in thermal tolerance. Populations selected under moderate, severe, and fluctuating warming all evolved increased tolerance of high temperatures relative to the ancestors and the control. Values are means and error bars denote ±1 s.e.m. Fitted lines are the fixed effects of a nonlinear mixed effects model. Gray curves denote the ancestor, green, samples are the control at 22 °C, blue at 26 °C, red at 32 °C, and purple is the fluctuating environment

however, lower than those for $R$ ($T_{opt}$ for $P$: 27 °C ± 0.7 °C, (±s.e. m.); $T_{opt}$ for $R$ = 29 °C ± 0.36 °C, (±s.e.m.)), meaning that above $T_{opt}$ for $P$, the carbon-use efficiency (CUE = 1−$R$/$P$; i.e., the potential carbon for allocation to growth) declined rapidly. The optimum temperature for growth rate ($T_{opt}$ = 28 °C; Fig. 1b) and subsequent decreases in growth at supra-optimal temperatures coincided with declines in the CUE above $T_{opt}$ for $P$ in the ancestor, demonstrating that the temperature dependence of CUE imposes a physiological constraint that shapes the thermal tolerance curve. Thus, we hypothesize that shifts in traits that increase CUE at high temperatures should have played an important role in facilitating adaptation to warming in the evolved lineages of *T. pseudonana* (see also Supplementary Table 9).

Following 300 generations of selection, we observed substantial shifts in the temperature responses of $P$ and $R$, both among treatments, as well as between the evolved lineages and the ancestor (Fig. 2; Supplementary Tables 6–9). For both $P$ and $R$, estimates of $T_{opt}$ in the warming treatments were higher than the ancestor and control, and rates of $P$ and $R$ declined less abruptly at temperatures exceeding $T_{opt}$. Consequently, CUEs remained high at hotter temperatures in the warming treatments. The mass-specific rates of $P$ and $R$ at a reference temperature $T_c$ = 18 °C (i.e., $P(T_c)$ and $R(T_c)$, in Eq. (6), which quantify the metabolic capacity per unit mass at a common temperature) were significantly down-regulated under moderate, severe, and fluctuating warming, relative to the control treatment and the

ancestor (Fig. 2c, d, Supplementary Tables 7–9). Rates of $R(T_c)$ also decreased more than those of $P(T_c)$, resulting in an overall increase in CUE in the lineages selected under moderate and fluctuating warming (Fig. 2e). These results support our hypothesis that fluctuating temperatures facilitate thermal adaptation and show that changes in the traits that increase CUE, played an important role in the evolution of elevated thermal tolerance in the treatments subjected to warming. It is notable, however, that the populations experiencing severe warming had lower growth rates and lower CUEs than the lineages in the moderate and fluctuating warming treatments (Figs 1b and 2e, Supplementary Tables 1 and 9, also Supplementary Figure 5). This suggests that while evolutionary rescue restored growth rates to levels comparable to the control, 300 generations of selection under constant, severe warming was insufficient to evolve CUEs that facilitate growth rates comparable to populations that experienced only brief periods of severe conditions.

**Contrasting effects of acclimation and evolution.** Changes in temperature are known to alter optimal subcellular allocation of resources to various macromolecular classes through acclimation (e.g., rapid changes in physiological traits not attributable to genetic change)[20–26]. Indeed, algae exposed to warming often increase their nitrogen-to-phosphorus ratios (N:P) by decreasing the density of P-rich ribosomes, relative to N-rich proteins, owing

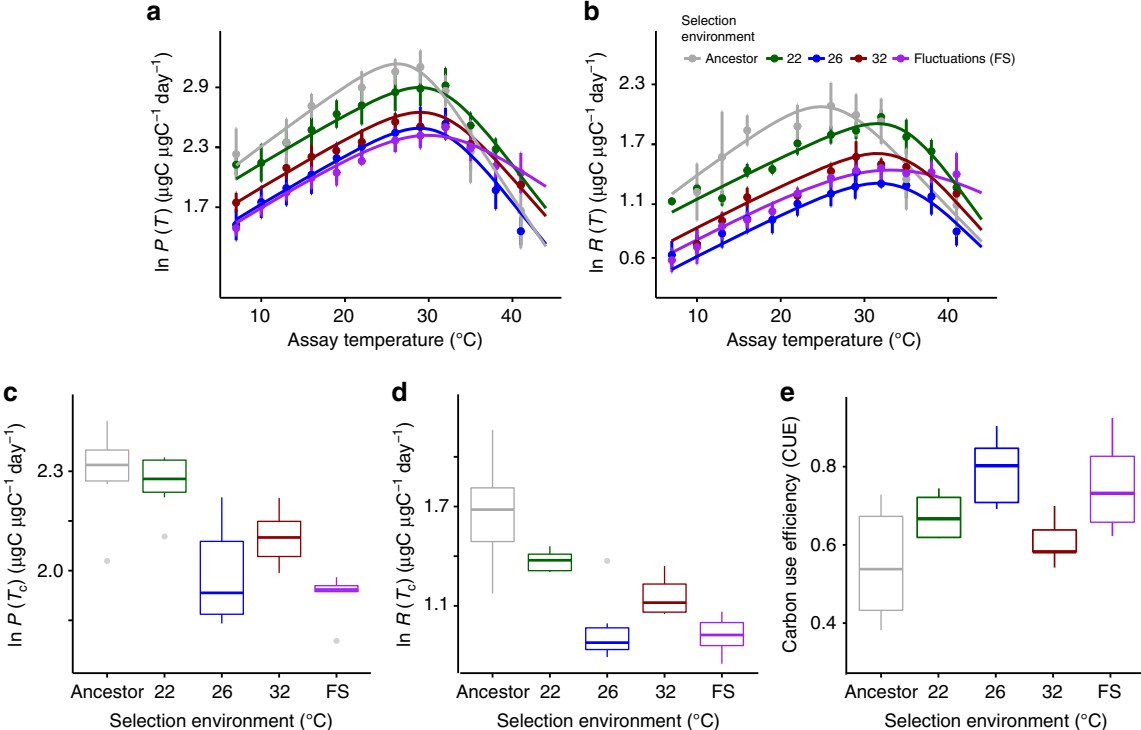

**Fig. 2** Rapid evolution of metabolic traits increase carbon-use efficiency (CUE). **a** Thermal response curves for gross photosynthesis rates ($P$), and **b** respiration ($R$) reveal substantial shifts in metabolic thermal traits among treatments, as well as between the evolved lineages and the ancestor. Data are presented as means with error bars denoting $\pm 1$ s.e.m, $n = 6$ per assay temperature, fitted lines are fixed effects from nonlinear mixed effects models. Mass-specific rates of **c** gross photosynthesis and **d** respiration at a reference temperature ($P(T_c)$ and $R(T_c)$ respectively, see Methods) were significantly down-regulated in the populations evolved under moderate, severe, and fluctuating warming. Furthermore, rates of $R(T_c)$ were down-regulated more than those of $P(T_c)$ in the moderate and fluctuating warming treatments meaning that **e** those lineages also had the highest CUE. Gray denotes the ancestor, green is the control at 22 °C, blue is at 26 °C, red is at 32 °C, and purple is the fluctuating environment (FS). CUE for FS is displayed at 32 °C for easier comparison to the 32 °C-evolved samples, and did not differ significantly between assay temperatures of 22 °C, 26 °C, and 32 °C (see also Supplementary Figure 5). Whiskers on boxplots indicate 1.5 × interquartile range

to the increased efficiency of protein synthesis by ribosomes at higher temperatures[23–26]. It is, however, unclear how these short-term changes in resource allocation are linked to the shape of the thermal tolerance curve (e.g., the optimum or upper thermal limit). To assess whether changes in subcellular allocation to various macromolecules could shed light on the processes that set the limits of thermal tolerance in the ancestor, we quantified how key traits, such as cell size, elemental composition, and photophysiology, changed through acclimation across the temperature gradient used to characterize the thermal tolerance curve of the ancestor (Fig. 1b). Consistent with expectations from the temperature-size rule[27], cell size declined linearly with increasing temperature (Fig. 3a, see also Supplementary Tables 10 and 11). In contrast, carbon (C), nitrogen (N), and phosphorus (P), expressed per unit cell volume to account for changes in cell size, all increased with temperature (Fig. 3b–d, see also Supplementary Tables 10 and 11). In line with the responses of N and P, protein and RNA content also increased with temperature (Fig. 3e, f, see also Supplementary Tables 10 and 11). The N:P and C:P ratios increased with rising temperature, but contrary to recent hypotheses[23–26], this was not driven by declines in P content, but rather, by more pronounced increases in N and C relative to P (Fig. 3h–j, see also Supplementary Tables 10 and 11). Notably, at temperatures exceeding the optimum for growth ($T_{opt} = 28$ °C), C, N, and P content, as well as protein and RNA content increased markedly. These findings are consistent with a greater demand for the transcription and translation of chaperones and other proteins required for the repair and maintenance of heat-induced damage. In further support for this hypothesis, the chlorophyll-to-carbon ratio showed a unimodal response to temperature, peaking at ca. 26 °C and declining rapidly thereafter (Fig. 3g, see also Supplementary Tables 10 and 11). Consequently, the increased N content observed at high temperatures cannot be explained by greater allocation to photosynthetic machinery. Instead, investment in proteins required for repair and maintenance provides a more parsimonious explanation for the high N content in cells exposed to supra-optimal temperatures.

To assess whether evolutionary adjustments in cell size and elemental composition could help to explain the evolution of elevated thermal tolerance, we quantified how these traits changed in the evolved lineages after 300 generations in their respective selection regimes. Cell size, C, N, and P content as well as protein and RNA content showed contrasting evolutionary responses to temperature changes relative to the short-term, acclimation responses. In the evolved lineages, cell size increased by a factor of almost 2 from 22 °C (control) to 32 °C (severe warming) (Fig. 3a, see also Supplementary Tables 10 and 11). This response contrasts with the linear decline in cell size observed as an acclimation response to short-term increases in temperature in the ancestor. Likewise, C, N, and P content as well as protein and RNA content, all declined between 22 and 32 °C in the evolved lineages (Fig. 3b–f, see also Supplementary Tables 10 and 11). Consistent with the short-term temperature responses in the ancestor, the N:P and C:P ratios both increased between 22 and 32 °C in the evolved lineages. However, in spite of the similarity in the overall temperature response, the changes in N:P

and C:P in the evolved lineages were driven by greater declines in P relative to N or C, consistent with expectations that fewer ribosomes are required at higher temperature owing to their greater per-ribosome catalytic capacity[23–26] (Fig. 3h–j, see also Supplementary Tables 10 and 11). This pattern contrasts directly with the short-term, acclimation response in which increases in N:P and C:P were driven by greater allocation to N and C with

rising temperatures. Thus, while the increases in N and P with rising temperatures are consistent with greater allocation to synthesizing new proteins to help combat increased costs of cellular maintenance in the short term, declines in N and P content following evolutionary adaptation may be indicative of re-allocation of metabolic energy back toward growth and biomass synthesis over the long term. Consistent with this, the

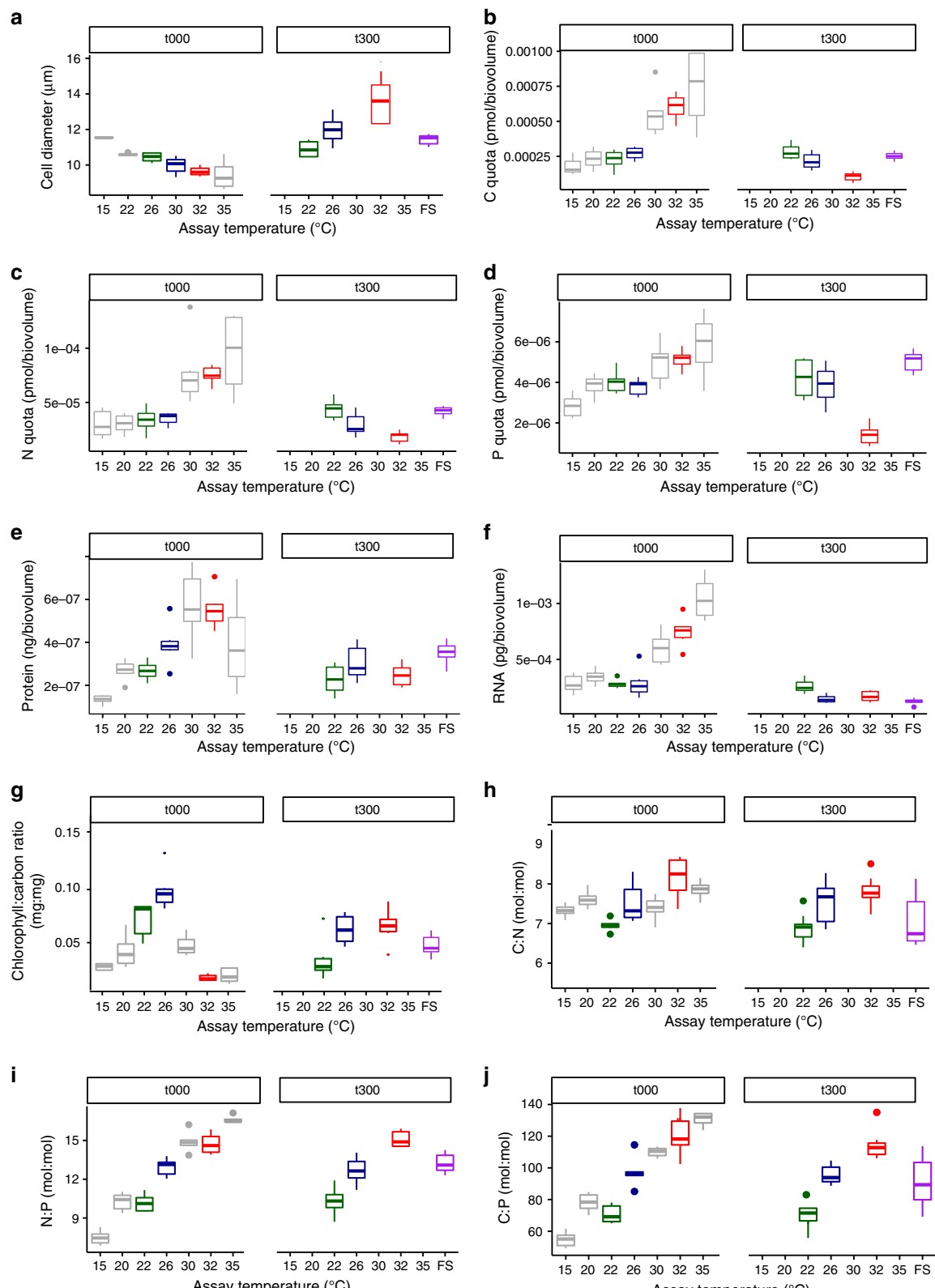

lineages that were selected under constant, severe warming (i.e., 32 °C) evolved substantially higher chlorophyll-to-carbon ratios relative to the short-term response of the ancestor when exposed to 32 °C, indicating that the evolved populations were re-allocating energy away from repair/maintenance and toward light harvesting machinery required for growth (Fig. 3g, see also Supplementary Tables 10 and 11). In general, the opposing effects of acclimation versus evolution on the direction of phenotypic trait responses to temperature change are consistent with geographic patterns of counter-gradient variation in which genetic influences on a trait oppose environmental influences thereby minimizing phenotypic change along the gradient[28].

**Molecular evolution of thermal tolerance.** To investigate whether the observed changes in fitness and phenotypes in the evolved lineages were also reflected by consistent patterns of molecular evolution, we re-sequenced the genomes of the ancestor and the mixed (non-clonal) populations of the evolved lineages after 300 generations of selection. We identified candidate single-nucleotide variants (SNVs) in the evolved populations compared to the ancestral clone that might have contributed to adaptation to the warming treatments. Specifically, two types of genetic adaptation were of interest: (i) mutations arising de novo after the $t_0$ time-point and becoming fixed in an evolved population and (ii) alleles present in the ancestral clone that become fixed in an evolved population. Fixation of de novo mutations would result in an allele proportion rising from zero in the ancestor to close to 1 after 300 generations of selection. Because the ancestor was a genetically homogeneous clone, standing variation consists solely as heterozygosity; we would therefore expect to observe alleles increasing in proportion from 0.5 in the ancestor to close to 1 after 300 generations. Therefore, we took two distinct approaches to surveying genetic variation: (i) we identified SNVs that were undetectable in the ancestral population but were supported by 100% of the aligned reads in an evolved population and (ii) we identified SNVs that plausibly had a population proportion of 0.5 in the ancestral population (due to heterozygosity), and were supported by 100% of the aligned reads in an evolved population.

We searched for non-silent SNVs affecting protein-coding genes that had become fixed in at least one of the evolved populations; such variants might have contributed to the populations' adaptation to the selection regime. SNVs were detected at 225,351 sites in the genome. Of these, 7383 had reached fixation in at least one evolved population, including 1689 missense variants and 14 gain-of-stop-codon variants. In total, 1215 protein-coding genes were affected by these fixed non-silent variants. Most fixed non-silent variants were present in the ancestor at proportions between zero and one and were therefore probably heterozygous in the ancestral clone (Fig. 4c), making the main mode of molecular evolution in these populations a transition from heterozygosity to homozygosity. Only nine of the fixed variants were undetectable in the ancestral clone, and

are thus likely to be de novo mutations (described in detail in the Supplementary Information). None of these nine de novo mutations were predicted to affect amino acid sequences of encoded proteins.

Using the SNVs that had reached fixation in at least one evolved population, we quantified the genetic distance of each population from the ancestor as well as the genetic divergence among populations using Analysis of Molecular Variance (AMOVA) based on squared Euclidean distances and visualized this analysis using Principal Components Analysis (PCA). Consistent with the fitness trajectories (Fig. 1a), evolved populations showed significant divergence from the ancestor and among one another (Fig. 4a; Table 1, Supplementary Table 12 for pairwise differences). The lineages selected under moderate and fluctuating warming had the greatest distance from the ancestor on the two major axes of variation (Fig. 4a; Supplementary Figure 6, Supplementary Table 12). Furthermore, the populations that experienced fluctuations between severe and benign conditions were divergent from those experiencing constant severe warming (Fig. 4a; Supplementary Figure 6 Supplementary Table 12), with significant variation in SNVs between treatments (Table 1, Supplementary Table 12). These results are consistent with the demographic data (Fig. 1a; Supplementary Figure 2), indicating that the temporary restoration of benign conditions in the fluctuating regime may have accelerated rates of molecular evolution by releasing cryptic genetic variation beneficial in the stressful environment. Selection lines in the moderate warming treatment and the fluctuating environment exhibited the greatest genetic divergence among replicate populations (Fig. 4a, Table 1—indicated by a larger $\sigma^2_{AP}$). Indeed, the genetic divergence among replicate populations was significantly higher in the moderate warming treatment compared to those exposed to stable, but severe warming (Fig. 4a, Table 1). The more conserved response in the severe environment may have plausibly arisen from stronger directional selection owing to the exposure to extreme conditions. The high genetic divergence among replicate populations in the fluctuating environment is consistent with expectations that more complex environments lead to greater diversity[29]. The above analyses were restricted to the subset of SNVs that had reached fixation in at least one of the evolved populations and for which we therefore we had a high degree of confidence that variants were either absent or heterozygous in the ancestor and rose to fixation in the evolved lineage. When considering all SNVs, regardless of their level of fixation, (see methods), the general patterns of genetic distance and divergence are qualitatively conserved (see Supplementary Figure 7), indicating that the results are robust to sequencing errors in estimating variants at low frequencies.

Comparing PCAs based on combining all the phenotypic data, with the PCA from the non-synonymous SNVs, revealed striking similarities in the patterns of divergence among treatments (Fig. 4a, b, Supplementary Figures 6 and 8). These results suggest that the observed changes in metabolic traits and elemental

**Fig. 3** Effects of thermal acclimation and adaptation on cell size and macromolecular composition. **a** Cell size (here as μm diameter) decreased with warming in the ancestor as an acclimation response, but cells became larger at warmer temperatures after 300 generations of evolution. Consequently, carbon (**b**), nitrogen (**c**), and phosphorus (**d**) quotas per cell volume of ancestral samples remained relatively stable up to the ancestor's optimum temperature and then increased, whereas elemental quotas in evolved lineages decreased with temperature (compare also Supplementary Figure 4 for per cell quotas). Protein (**e**) and **f** RNA content per unit cell volume mirrored the changes in intracellular N and P content respectively. The chlorophyll-to-carbon ratio was also highest in the fastest growing lineages selected under moderate warming (**g**) and the C:N (**h**), N:P (**i**), and C:P (**j**) ratios increased with increasing selection temperature both in the short- and in the long-term, with values in the fluctuating environment comparable to those in the moderate warming treatment – See Supplementary Information for $\Phi_{PSII}$ (Supplementary Figure 3) and silicate content (Supplementary Figure 4). For all panels, $n = 6$. Boxplots are for the ancestor (gray) and evolved lineages after 300 generations at 22 °C (green), 26 °C (blue), 32 °C (red) and the fluctuating environment (purple). Whiskers on boxplots indicate 1.5 × interquartile range

composition that were linked to the evolution of thermal tolerance could also have a basis in some of the underlying genomic changes. To investigate this, we identified the top 100 SNVs most strongly associated with the first and second principal components as these are expected to be the most important SNVs for characterizing genomic variation among the lineages. We then identified the genes (see Supplementary Table 14) in which these

SNVs fell and tested whether this gene set was enriched in any Gene Ontology (GO)[30,31] terms related to biological processes, using the PCA loading score to weight the relative importance of each gene. Genes that had gone to fixation in the evolved lineages and were most strongly associated with the major axes of variation in the PCA were enriched in GO terms related to "transcriptional regulation", "cellular response to oxidative stress"

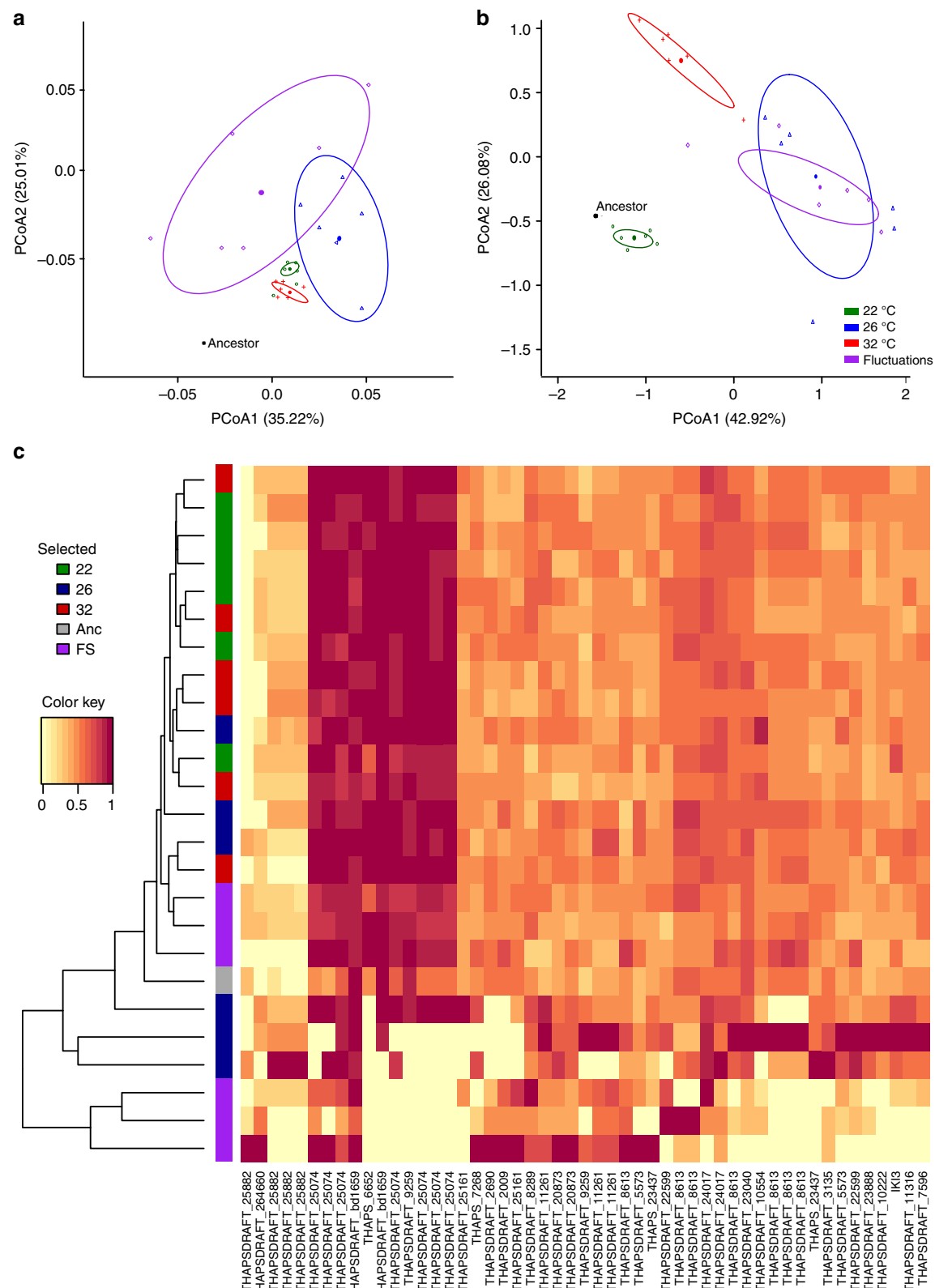

and "redox homeostasis" among others (Table 2). Notably, the genes that correspond to these enriched GO terms were all most strongly associated with the moderate and fluctuating warming treatments in the PCA.

We also found that several populations from the moderate and fluctuating warming treatments displayed fixation of exceptionally high numbers of SNVs (Supplementary Figures 12 and 13). In these lineages, the ancestral population's heterozygosity was lost at hundreds of single-nucleotide sites in the genome. The mechanism(s) of heterozygosity loss among these populations is not known; however, based on analysis of sequencing depth, we also observed substantial copy-number variation among populations across several large genomic regions (Supplementary Figures 9–11). These findings are consistent with similar analyses of the *Chlamydomonas reinhardtii* genome in laboratory and wild-type isolates[32]. This suggests that large-scale duplications and deletions might be important mechanism for rapid evolution in algae, for example, when recurrent copy-number gains represent adaptive gain-of-function mutations[33], dosage effects[34], and deletions harbor potentially non-sense mutations, such as premature stop codons.

## Discussion

Our results shed new light on the adaptive potential and evolutionary dynamics of one of the most abundant and widely distributed eukaryotic marine phytoplankton in response to warming. We found that evolutionary rescue under severe warming was slow (>1 year), but adaptation to more realistic warming scenarios, where temperature increases were moderate or where they fluctuated between benign and severe conditions, was rapid. The fluctuating environment accelerated adaptation to severe warming because temporary restoration of benign conditions increased population size and therefore the probability of fixing beneficial mutations required for adaptation to the severe environment. Consistent with this demographic effect, the lineages selected under fluctuating warming had the greatest genetic distance from the ancestor after 300 generations, indicating accelerated rates of molecular evolution. Because our experiments were initiated with a single clone, they may in fact be conservative estimates of the evolutionary potential of the highly diverse meta-population of *T. pseudonana* in the wild[5,35] where adaptation will also be aided by standing genetic variation. Furthermore, our experiments were conducted on a clone of the CCMP 1335 strain of *T. pseudonana*, which has been in laboratory culture for many decades. Consequently, this strain has likely lost some of its genomic diversity owing to long-term adaptation to simplified laboratory conditions, further increasing the likelihood that observed evolutionary responses observed in our experiments are conservative estimates. Similar experiments on freshly isolated strains and diverse phytoplankton taxa are urgently needed to provide improved estimates of the evolutionary potential of marine phytoplankton to adapt to global change. Our results also show how rapid, adaptive shifts in

thermal tolerance are linked to major phenotypic changes in metabolism and elemental composition as well as significant molecular evolution, particularly in the populations evolved under moderate and fluctuating warming. Because the changes in CUE in the warm-adapted lineages arise due to differences the temperature sensitivities of photosynthesis and respiration, which are highly conserved metabolic pathways, our work raises the prospect that at the physiological-level, the mechanisms underpinning rapid thermal adaptation could be universal across the broad diversity of phytoplankton. Our findings could therefore help integrate rapid evolution into models of ocean biogeochemistry and improve projections of marine primary productivity over the 21st century[1,36].

## Methods

**Experimental design.** In November 2014, a clone of the sequenced strain of *T. pseudonana* (CCMP 1335)[11] was obtained from the CCAP culture collection, where it had been re-isolated from the original clone CCMP 1335 in 2009, and maintained in batch culture at 18–22 °C. This lineage is known to exhibit a unimodal growth response to temperature between 16 and 31 °C, with optimal growth temperatures around 27 °C[37–39]. The stock culture was made clonal by serial dilution in a 96-well plate in March 2015 and a single clone was chosen at random to be the ancestor for all treatments in the selection experiment. Determination of the thermal tolerance curve of the ancestor revealed that growth rates peaked at 28 °C and that mortality was high above 35 °C (see section below for details on characterizing the thermal tolerance curves). The ancestor was distributed among 4 experimental treatments that represent a range of warming scenarios based on its limits of thermal tolerance: (i) a control at 22 °C, which was the long-term culture temperature; (ii) moderate warming at 26 °C; (iii) severe warming at 32 °C and (iv) a fluctuating thermal regime, which cycled between 22 and 32 °C every 3–4 generations. Each treatment was replicated 6 times.

Lineages were grown in f/2 Medium (Guillard's medium for diatoms[40]) with artificial seawater, under a 12:12 light/dark cycle in INFORS HT shaking incubators. Salinity was maintained at 32 (i.e., 32 g NaCl L$^{-1}$ in 39.5 g L$^{-1}$ artificial seawater reagents) and pH adjusted to 8.213 ± 0.291 (±s.e.m., averaged across all transfers) prior to each transfer. pH was further measured at the end of each transfer (average across all transfers: 8.171 ± 0.252). The light intensity of full-spectrum fluorescent tubes was 100 μmol quanta m$^{-2}$ s$^{-1}$. As light intensity varied throughout the incubators (minimum light intensity 70, maximum light intensity 110 μmol quanta m$^{-2}$ s$^{-1}$), culture vessels were moved randomly within the incubator every other day. Lineages were maintained in semi-continuous batch culture, and transferred during the exponential phase of growth to an inoculum of 100 cells mL$^{-1}$. Average cell densities were calculated at each transfer from three technical replicates for each biological replicate.

**Flow cytometry.** Abundance and cell size (diameter assuming a spherical shape) were determined with a Accuri C6 flow cytometer (BD Scientific). Cell size was detected using forward scatter calibrated (FSC) using species and beads of known size against photographic images processed in ImageJ as described in ref.[41], with a conversion factor of FSC = 108058 μm + 4.5665. Although *T. pseudonana* cells are cylindrical rather than spherical, this is a useful approximation for relative changes in cell size. Red fluorescence (FL3) was used for chlorophyll fluorescence in order to distinguish autotrophs from bacteria, but was not used here to approximate chlorophyll content, which was instead estimated through acetone extraction (see below). Co-occuring bacteria, e.g., *Rhodobacter* spp. and *Marinobacter* spp. are known to be crucial for nutrient recycling and signaling molecule metabolism in *T. pseudonana*[42,43]. We therefore made no attempt to make the cultures axenic as this would have incurred a substantial fitness cost. We quantified the associated bacteria by regularly filtering the cultures between 0.2 and 2 μm, staining with SYBR gold and enumerating bacterial densities and cell size via flow cytometry. Co-occurring bacteria contributed less than 1% of total biomass throughout, with no systematic variation between time-points or treatments.

**Fig. 4** Molecular and phenotypic evolution in *T. pseudonana*. **a** Principal component analysis (PCA) of evolved and ancestral lineages calculated from distance matrices based on single-nucleotide variants (SNVs) in protein-coding regions that had reached fixation after 300 generations of evolution. **b** PCA of phenotypic traits calculated from the change in each trait measured at the treatment temperature relative to that expressed in the ancestor at the same temperature. In panels **a**, **b** colors indicate the selection regime with black for the ancestor, green for 22 °C, blue for 26 °C, red for 32 °C, and purple for the fluctuating regime. The ellipses in PCAs give the 95% confidence interval around the centroid and thus give an idea of whether the treatments differ significantly (detailed information on the statistical significance of separation of the selection environments can be found in Table 1 & Supplementary Tables 12 and 13). **c** Estimated allele frequencies for 1703 single-nucleotide variants that putatively reached fixation in at least one t300 population. Of these, 1689 are missense, and 14, stop-codon variants. The color of each cell in the heat-map indicates the estimated allele proportion in the population, based on the ratio of variant sequence reads versus total read depth at that genomic site. Homozygously fixed alleles will appear as yellow or red (allele proportion zero or one, respectively), heterozygously fixed alleles as orange. The side-bar indicates the selection regime with colors as stated above

**Table 1 Results of the analysis of molecular variance (AMOVA). AMOVA was used to assess genetic divergence within treatments (i.e., among biological replicates in a given treatment) and between treatments (i.e., between the different selection environments)**

|  | $\sigma^2$ | % | *p* value |
|---|---|---|---|
| **22 °C** |  |  |  |
| Within population ($\sigma^2_{WP}$) | 39.76 | 74.99 | <0.05 |
| Among population ($\sigma^2_{AP}$) | 13.26 | 25.01 | <0.05 |
| Total ($\sigma^2_T$) | 53.02 | 100 |  |
| **26 °C** |  |  |  |
| Within population ($\sigma^2_{WP}$) | 31.07 | 52.29 | <0.01 |
| Among population ($\sigma^2_{AP}$) | 28.35 | 47.71 | <0.01 |
| Total ($\sigma^2_T$) | 59.42 | 100 |  |
| **32 °C** |  |  |  |
| Within population ($\sigma^2_{WP}$) | 43.66 | 86.51 | <0.05 |
| Among population ($\sigma^2_{AP}$) | 6.81 | 13.49 | 0.051 |
| Total ($\sigma^2_T$) | 50.47 | 100 |  |
| **FS** |  |  |  |
| Within population ($\sigma^2_{WP}$) | 31.03 | 50.04 | <0.01 |
| Among population ($\sigma^2_{AP}$) | 30.98 | 49.96 | <0.01 |
| Total ($\sigma^2_T$) | 62.01 | 100 |  |
| **All treatments** |  |  |  |
| Within population ($\sigma^2_{WP}$) | 28.97 | 49.69 | <0.05 |
| Among populations between treatments ($\sigma^2_{AP/BT}$) | 8.82 | 15.12 | <0.05 |
| **Between treatments ($\sigma^2_{BT}$)** | **20.522** | **35.19** | **<0.05** |
| Total ($\sigma^2_T$) | 58.32 | 100 |  |

AMOVA revealed that a significant proportion of the variance in SNV frequencies at specific loci was attributable to differences between treatments. Within individual treatments, there was significant variation between biological replicates ("populations" in the table), with the largest amount of variation among populations, ("AP"), in the 26 °C and FS treatments. $\sigma^2$ represents the variance of each hierarchical level (i.e., within population "WP", among populations "AP" and between treatments "BT"), and the % gives the percentage of this variance calculated to a total of 100%. Significance testing was carried out through random permutation of the samples following the methods outlined in ref. [59]

**Table 2 Table of Gene Ontology (GO) terms. GO terms relating to biological processes that were significantly enriched in the gene set associated with the major axes of variation in the principal components analysis**

| GO.ID | Term | Ann | Sig | Exp | KS | Gene | Loading # | Comp | Treat | Loading score |
|---|---|---|---|---|---|---|---|---|---|---|
| GO:1902358 | Sulfate transmembrane transport | 12 | 1 | 0.02 | 0.015 * | THAPS_23437 (hypothetical protein) | 2 | comp1 | FS | 0.0559 |
| GO:0006357 | Regulation of transcription from RNA polymerase | 44 | 1 | 0.03 | 0.015 * | THAPSDRAFT_7370 (hypothetical protein) | 27 | comp1 | FS | 0.0483 |
| GO:0001522 | Pseudouridine synthesis | 44 | 1 | 0.03 | 0.025 * | THAPS_23513 (hypothetical protein) | 40 | comp1 | 26 | 0.0472 |
| GO:0034599 | Cellular response to oxidative stress | 23 | 1 | 0.03 | 0.029 * | THAPSDRAFT_25121 (hypothetical protein) | 22 | comp2 | FS | 0.0570 |
| GO:0008152 | Metabolic processes | 77 | 1 | 0.03 | 0.035 * | THAPSDRAFT_23040 (hypothetical protein) | 6 | comp1 | 26 | 0.0533 |
| GO:0045454 | Cell redox homeostasis | 77 | 1 | 0.03 | 0.035 * | THAPSDRAFT_2086 (hypothetical protein) | 31 | comp1 | 26 | 0.0478 |
| GO:0000103 | Sulfate assimilation | 29 | 1 | 0.02 | 0.037 * | THAPSDRAFT_25121.2 (hypothetical protein) | 22 | comp2 | FS | 0.0570 |
| GO:0016567 | Protein ubiquitination | 3706 | 5 | 4.75 | 0.254 | THAPS_23513,THAPSDRAFT_2086,THAPSDRAFT_7370 (hypothetical protein) | Various | comp1 and comp2 | FS and 26 | Various |

GO.ID is the Gene Ontology identifier retrieved from protists.ensembl.org, *term* is the biological process associated with the GO.ID, and *Ann* is for the number of genes annotated to the GO.ID category found in the dataset of genes that are associated with the top 100 PCA loadings for the two principal components displayed in Fig. 4a. *Sig* and *Exp* denote the number of significant and expected annotations for the GO.ID category found in the dataset of genes that are associated with the top 100 PCA loadings for the two principal components displayed in Fig. 4a compared to the reference "gene universe" made up from the entire *T. pseudonana* genome. *KS* is the *p* value output of a Kolmogorov–Smirnov test, which replaces Fisher's exact test when working with scores (see Methods) with *p* < 0.05 indicated by * for significant enrichment. Loading # indicates the position that the gene has in the PCA's top 100 loadings, and comp indicates whether it is more strongly associated with the first or second principal component (see Fig. 4a). The loading score is a numerical value for the strength of the association with a component, where higher absolute values are indicative of a stronger association. Gene gives the locus tag of the gene that was found to be strongly associated with the axis of variation. All genes are coding for hypothetical proteins, and were assigned GO terms through the most likely function that the protein may have given its amino acid sequence. *Treat* denotes the treatment that SNVs in that gene were most likely to be associated with based on the PCA

Flow cytometry was further used in combination with a rhodamine stain to measure a proxy for mitochondrial potential—the rate of ATP production by ATP synthase is proportional to mitochondrial potential, which in turn depends on the activity of respiratory proteins embedded in the mitochondrial membrane. Since each unit area of membrane has several ATP synthase molecules and the rate at which each of these produces ATP depends on the mitochondrial potential, the overall rate of ATP production per mitochondria depends on both the area of mitochondrial membrane and the mitochondrial potential. Mitochondrial area is a good approximation of the area of mitochondrial membrane, and changes in mitochondrial potential can be measured in live cells using potentiometric dye. Previous work[44] has shown that the styrl dye rhodamine 123 (R123) can be used in live cells of phytoplankton to measure mitochondrial potential in real time. For staining, a 10 mg mL$^{-1}$ stock solution in DMSO was prepared and diluted to a 1 μM working solution on the day of measurements. Of this, 20 μL were added to 200 μL of sample and left to incubate in the dark for 45 min. To ensure that longer incubation as is inevitable on a 96 well plate did not effect FL1 fluorescence as measured in the accuri c6, samples were randomized across each 96-well plates with the first and the last well containing the same sample. We found no effect of prolonged incubation after an incubation period of 45 min.

A Nile Red stain was performed to measure relative lipid contents. Here, the dye was used as a proxy to determine relative quantities of intracellular polar and neutral lipids[45] in evolved samples, and in samples from short-term assays. The stock dye was obtained as a powder. The working solution was diluted to 0.1 mg L$^{-1}$, and 10 μL were added to each 200 μL sample on a 9-well plate (final concentration 15 mM) and left to incubate in the dark for 30 min, as pilot trials had shown that after this, fluorescence levels were stable long enough for the time taken to measure one 96 well plate. Samples were randomized on the plate as described above for R123 stains. As Nile Red excites in the same wavelength as chlorophyll (FL3) and chlorophyll derivatives (FL2), samples were measured before and after adding the dye, and the chlorophyll fluorescence subtracted from the fluorescence obtained after staining the sample. While these traits are not discussed in detail here, they are part of the distance matrices used to generate Fig. 4b (see also Supplementary Figure 8 and Supplementary Table 13).

**Growth rate trajectories**. At the beginning and at the end of each transfer *T. pseudonana* cells were counted on the flow cytometer as described above and used to estimate specific growth rates ($\mu$, day$^{-1}$)

$$\mu = \frac{\ln(N_{t1}) - \ln(N_{t0})}{\Delta t} \qquad (1)$$

where $N_{t1}$ is the density of cells at the end of the transfer, $N_{t0}$ is the inoculation density, and $\Delta t$ is the time passed in days.

**Thermal tolerance curves**. To characterize the thermal tolerance curves of the ancestor and each of the evolved lineages, an inoculum of 100 cells per mL from the middle of the logarithmic phase of growth was transferred into fresh media at 15, 20, 25, 30, 32, 35, and 40 °C. Cell count was determined daily on a flow cytometer and populations were transferred to fresh media once at each temperature during the middle of the logarithmic phase of growth, before being left to reach stationary phase. Growth rates were estimated from the abundance data using a logistic growth function:

$$N(t) = \frac{K}{1 + \frac{K - N_{t0}}{N_{t0}} e^{rt}} \qquad (2)$$

where $t$ is time, $K$ is carrying capacity of the population (cell mL$^{-1}$), $r$ is the maximum growth rate (day$^{-1}$), and $N_{t0}$ is the cell count at $t_0$ (cell mL$^{-1}$).

**Intracellular macromolecular composition**. Cellular carbon (C), nitrogen (N), phosphorus (P), and chlorophyll content (total chlorophyll calculated from chlorophyll *a* and chlorophyll *c* content) were quantified in the ancestor and each of the evolved lineages. We investigated the effects of temperature on macromolecular composition both over short-term acclimation and long-term evolutionary adaptation. Short-term thermal acclimation was investigated by exposing the evolved lineages to the 15–45 °C thermal gradient described above for quantifying the thermal tolerance curves for growth rate. Samples from the 40 and 45 °C environment were omitted from further analyses due to high mortalities that limited the amount of biomass available for further tests. Lineages were given 1 transfer to acclimate to a given assay temperature in the gradient and then harvested during the logarithmic phase of growth in the second transfer. Long-term evolutionary adaptation to the selection regimes was quantified by harvesting evolved lineages directly from the experiment during the logarithmic phase of growth

The remaining samples were prepared for C, N, P, and chlorophyll measurement by spinning down 50 mL of culture in a centrifuge at 4 °C and 3500 RPM (rounds per minute) for 30 min. The pellet was then transferred to a 1 mL Eppendorf tube, spun again for 15 min at 3500 RPM and the remaining supernatant decanted. Chlorophyll content was determined on a spectrophotometer (Jenway 7351) after[46], with extraction in acetone, and absorption spectra measured in 10 nm increments from 500 to 700 nm, which span the full range of emission peaks for chlorophyll *a* and chlorophyll *c*.

Pellets for C and N determination were freeze dried for 24 h, transferred to zinc capsules, weighed and analyzed using a SerCon Isotope Ratio Mass Spectrometer (CF-IRMS) system (continuous flow mode). Intracellular P content was determined via a colourimetric reaction on a Seal Analytics AA3 segmented flow auto-analyzer. Freeze dried pellets were washed in 0.17 M Na$_2$SO$_4$, transferred to scintillation vials and resuspended in 4 ml 0.017 M MnSO$_4$. The samples were transferred to an autoclave (1 h, 121 °C), shaken vigorously and centrifuged at 2500 r.p.m. for 30 min, the pellet discarded, and the supernatant brought to 10 ml with MillliQ purified water. The samples were immediately analyzed on the AA3 using the colourimetric molybdate/antimony method[47]. Using these data we calculated the molar stoichiometric ratios, C:N, C:P, N:P, and the Chl:C ratio.

Silica content, although not a main trait under investigation here, was determined through preparing pellets as described above. Then, 4 ml of 0.2 M NaOH were added to the pellet in polyethylene tubes and the pellets vortexed briefly. The samples were then transferred to a heating block at 90 °C for ca. 1 h. Two milliliters of 1 M HCl were added, the samples vortexed again and then spun at 3500 rpm for 1 h. The supernatant was passed through an 0.5 µm filter, pipetted into fresh tubes, topped up to 10 ml with MiliQ water and used immediately on the segmented flow auto-analyzer, where the colourimetric procedure followed that described in ref. [48].

**Total RNA and total protein content**. 100 mL of sample in exponential growth were condensed to a pellet via centrifugation at 4 °C and 3500 RPM (rounds per minute), and were snap-frozen in liquid nitrogen until further use. Total RNA was extracted using a Qiagen RNA isolation kit and purified following the Qiagen RNeasy protocol with an in-column DNAse treatment, as per the supplier's instructions. Total RNA content in µg/mL was measured immediately afterwards on a Qubit system and on an Agilent BioAnalyser. Total protein concentrations of snap-frozen cell pellets were obtained from samples in exponential growth through extraction on ice following the protocol of ref. [49]. The frozen cell pellet was placed on ice, and re-suspended with 200 µL buffer containing 50 mM MOPS, 0.05% Triton™ X-100, 10 mM DTT and a protease inhibitor cocktail (Sigma). To achieve

lysis, cells were sonicated for 30 minutes on ice and the lysate was centrifuged at 4 °C and 3500 RPM for 30 minutes. The supernatant was transferred into a clean tube, proteins precipitated in cold acetone on ice at −20 °C overnight, and centrifuged again at 4 °C and 3500 RMP for 10 minutes the following morning. The supernatant was discarded and the pellet was dissolved in 100 µL TEAB (Triethylammonium bicarbonate buffer) buffer with 0.1% Triton™ X-100. Where necessary, sonication steps were repeated until the samples were completely solubilised. Finally, samples were reduced with TCEP (Tris (2-carboxyethyl) phosphine hydrochloride) and any remaining insoluble material removed through centrifugation. Total protein concentration was then measured using a colorimetric method (Bradford reagent, Bio-Rad), according to the supplier's protocol (see also ref. [49]). This was followed by trypsin digestion. Digested samples were stored frozen until further use (e.g. visualisation using an Agilent BioAnalyser). Values reported here are for *total* protein per cell, prior to digestion, and are not indicative of peptide length or protein composition.

**Photochemistry**. We characterized a range of photochemical parameters in the ancestor and each of the evolved lineages using fast repetition rate fluorometry (FastPro8, FRRf3, Fast Ocean System Chelsea Technology Group). 500 µL of dilute sample (with a cell count of less than 1000 cells mL$^{-1}$) were added to 5 ml of fresh culture medium. Samples were then pre-incubated in the dark at the assay temperature for 15 min in a water-bath, and another 10 min in the fluorometer to make sure that samples were fully dark acclimated and all reaction centers closed. All measurements were made at the selection temperature for the 22, 26, and 32 °C lineages, while for the lineages in the fluctuating regime measurements were carried out at both 22 and 32 °C. Photochemical traits were measured in response to rapid flashes at increasing light intensities from 0 to 1600 µmol m$^{-2}$ s$^{-1}$. Flash frequency and rate followed standard protocols for phytoplankton[50], with 100 flashes of 1.1 µs at 1 µs intervals. Peak emission wavelengths of the LEDs used for excitations were at 450, 530, and 624 nm. $\Phi_{PSII}$ was particularly relevant to our study as it is commonly used to describe the light responses of photosynthetic efficiency. $\Phi_{PSII}$ values are used as an indication of the proportion of the total light absorbed that is used in photochemical reactions in PSII (Supplementary Figure 3, Supplementary Table 16). Additionally, we determined rP, NPQ, and $C$ (see Fig. 4b and Supplementary Table 13 for PCA on phenotypic traits). These describes the relative rate of photosynthesis in response to irradiance and are obtained as an estimate of electron transport through PSII (rP), the cell's ability to maintain photochemical function at high light intensities (non-photochemical quenching—NPQ) and the proportion of PSII reaction centers in a closed state ($C$). As measurements were carried out following a light response curve, we were then able to measure these functions both at saturating light intensity and at the light intensity that the samples were grown at in the incubators.

**Thermal responses of photosynthesis and respiration**. Measurements of photosynthesis and respiration were made on the ancestor and all evolved lineages after 300 generations of selection when in the middle of the logarithmic phase of population growth. Net photosynthesis (NP) was measured as O$_2$ evolution at increasing light intensities in intervals of 50 µmol$^{-1}$ m$^{-2}$ s$^{-1}$ up to 300 µmol$^{-1}$ m$^{-2}$ s$^{-1}$, and then in intervals of 100 µmol$^{-1}$ m$^{-2}$ s$^{-1}$ up to 1000 µmol$^{-1}$ m$^{-2}$ s$^{-1}$, followed by 200 µmol steps up to 2000 µmol$^{-1}$ m$^{-2}$ s$^{-1}$. The maximum rate of light saturated photosynthesis was determined by fitting the NP data to a dynamic model of photoinhibition via nonlinear least squares regression using the methods outlined in ref. [19]:

$$NP(I) = \frac{NP_{max}I}{\frac{NP_{max}}{\alpha I_{opt}^2}I^2 + \left(1 - 2\frac{NP_{max}}{\alpha I_{opt}}\right)I + \frac{NP_{max}}{\alpha}} - R \qquad (3)$$

where NP($I$) is the rate of photosynthesis at light intensity $I$, NP$_{max}$ is the maximum rate of net photosynthesis at the optimal light intensity, $I_{opt}$, and $\alpha$ controls the rate at which NP($I$) increases up to NP$_{max}$, and $R$ is the rate of respiration (i.e., the rate of O$_2$ flux when $I = 0$). Gross photosynthesis ($P$) was estimated as $P = NP_{max} + R$. Measurements of O$_2$ flux were made in a Clark-type oxygen electrode (Hansatech Ltd, King's Lynn UK Chlorolab2). Aliquots (50 ml) of the populations were concentrated through centrifugation to a density of approximately $8 \times 10^4$ cells mL$^{-1}$ and acclimatized to the assay temperature for 15 min in the dark before measuring metabolic rates. The thermal responses of $P$ and $R$ were quantified by measuring rates over a temperature gradient from 4 to 45 °C (in 3 °C increments, with additional measurements at the selection temperatures). Rates of $P$ and $R$ were expressed in units of µg C per µg C using the following equation[51]:

$$b(\mu g C\ \mu g C^{-1}) = \frac{b\left(\mu mol\ O_2\ mL^{-1}\ day^{-1} * 32 * M * \left(\frac{12}{44}\right)\right)}{\mu mol\ C\ cell^{-1} * cells\ mL^{-1}} \qquad (4)$$

where $b$ is the metabolic rate (either $P$ or $R$), 32 * $M$ * (12/44) is used to convert µmol O$_2$ into µg C, and the factor $M$, is the assimilation quotient of CO$_2$:O$_2$. In our study, NO$_3^-$ was the source of nitrogen, thus assuming a set of balanced growth equations[51]

$$nCO_2 + (n+1)H_2O + HNO_3 \rightarrow (CH_2O)_n\,NH_3 + (n+2)O_2 \qquad (5)$$

if the C:N ratio is calculated in moles the assimilation ratio of $CO_2:O_2$ will be $n/n$ +2 (see ref.[51]). The estimated $M$ values are given in Supplementary Table 1 and ranged from ~ 0.77 to ~ 0.79. Alongside the biomass measurements for phytoplankton, we also periodically measured bacterial biomass (see methods detailed above), which typically comprised <1% of algal biomass. Consequently, bacterial respiration did not contribute significantly to metabolic rates, and when 2 μm filtered samples were run for metabolic rate measurements, the slopes were indistinguishable from water blanks.

**DNA extraction for whole-genome re-sequencing**. At the beginning (t0) and after 300 generations (t300) of selection in each environment, 250 ml of each sample in exponential growth were spun to a pellet in 50 ml batches at 4 °C and 3500 RPM. The supernatant was discarded and the pellet stored at −20 °C until further use. Prior to DNA extraction, samples were put through three cycles of thawing and refreezing, as this had shown to increase yield of diatom DNA relative to bacterial DNA in pilot studies. DNA was extracted following a standard cTAB protocol in chloroform:isoamyl and isopropanol[52] with a proteinase K step to digest proteins, and an RNAse step to digest RNA. The resulting DNA pellet was quality controlled on 1% agarose gels and fluorometrically on a Qbit system. The extracted genomic DNA was then frozen in TE buffer at −20 °C until samples were processed at the Exeter Sequencing Services (University of Exeter, UK). Whole-genome re-sequencing (Illumina) was carried out on a HiSeq 2500 platform with paired-end reads. Library preparation was carried out at the Exeter Sequencing Services using Netflex. The resulting reads were assembled against the existing *T. pseudonana* genome[11].

**Alignment of sequence reads against the reference genome**. A prerequisite of variant calling from genomic shotgun sequencing data is the alignment of sequence reads against a reference sequence[45], in this case the genome sequence of *T. pseudonana* (GenBank: GCA_000149405.2). Prior to alignment, reads were trimmed and filtered to remove low-confidence base-calls by applying TrimGalore with command line options "-q 30" and "--paired". Alignment was performed using BWA-mem version 0.7.12-r1039[53,54]. As a precaution against artifacts arising from non-uniquely mapping reads we used command line option "-q 1" when using samtools view[54] to convert the alignments from SAM to BAM format; use of this option has the effect of excluding from the alignment all sequence reads that map equally well to more than one unique site on the reference genome sequence.

**Calling single-nucleotide variants**. We identified candidate SNVs in BWA-mem alignments of genomic sequence reads against the reference genome sequence using the SAMtools/BCFtools pipeline version 1.3.1[54]. Candidate SNVs were identified using bcftools 1.3.1 with the following command-lines:
"samtools mpileup -u -f genome.fasta alignment.bam >alignment.bcf" and
"bcftools call -m -v –Ov alignment.bcf >alignment.vcf".
The candidate variants were then filtered using the following command line: "bcftools filter --SnpGap 10 --include "(REF = "A" | REF = "C" | REF = "G" | REF = "T") & %QUAL >= 35 & MIN(IDV) >= 2 & MIN(DP) >= 5 & INDEL = 0" alignment.vcf >alignment.filtered.vcf". This filtering step eliminates indels with low-confidence SNV calls. It also eliminates candidate SNVs within 10 base pairs of an indel, since alignment artifacts are relatively common in the close vicinity of indels. Finally, annotation was added to each filtered VCF file using snpEff[55] version 4.3t.

The motivation for variant calling was to identify candidate genetic changes in the evolved populations (compared to the ancestral clone) that might contribute to thermal adaptation. Specifically, two types of genetic adaptation were of interest: (i) mutations arising de novo after the $t_0$ time-point and becoming fixed in an evolved population and (ii) alleles present in the ancestral clone, that become fixed in an evolved population. In both cases, fixation could arise through either drift or selection. Fixation of de novo mutations would result in allele proportion rising from zero at $t_0$ to close to 1 by 300 generations. Because the ancestral population was a genetically homogeneous clone, standing variation consists solely as heterozygosity, and we expect to observe alleles increasing in proportion from 0.5 at $t_0$ to close to one by 300 generations. Therefore, we took two distinct approaches to surveying variation: (i) identifying SNVs that were undetectable in the ancestral population (at a sequencing depth of at least 10×) but were supported by 100% of the aligned reads in an evolved population (at depth of at least 5×) and (ii) identifying SNVs that plausibly had a population proportion of 0.5 in the ancestral population (due to heterozygosity), at depth of at least 10×, and were supported by 100% of the aligned reads in an evolved population, at depth of at least 5× (see also Supplementary Table 15 for average sequencing depths).

The output of SNV-calling was a matrix of estimated allele frequencies at each SNV site for each sample. All SNVs included in the matrices had passed reliability filtering, as described above. Following the approach of a recent study[50], the estimated allele frequency was defined as the ratio of sequence reads matching the alternative allele versus total number of reads aligned at that site. In other words, the sample proportion is used as an estimate of the proportion:

$$f_{pm} \approx \frac{A_{pm}}{D_{pm}}$$

where for mutation $m$ and population $p$, $A_{pm}$ and $D_{pm}$ are, respectively, the count of aligned reads matching the alternate allele and the total counts (i.e., depth). Inevitably, this estimator of $f_{pm}$ will be confounded by sampling error. Assuming that each sequence read is drawn randomly from the population, the sample proportion will follow a binomial distribution:

$$A_{pm} \sim \text{Binomial}(D_{pm}, f_{pm})$$

Given the sample size (in this case, $D_{pm}$) and the population proportion ($f_{pm}$), it is possible to calculate the 95% confidence interval of the estimate of $f_{pm}$[56]. With small sample sizes, the confidence intervals are wide. For example, with $D_{pm}$ (depth) of 10× and $f_{pm}$ (allele frequency) of 0.5, the 95% confidence interval is 0.5 ± 0.31. For depth 50× it is 0.5 ± 0.14 and for depth 100× it is 0.5 ± 0.10 (see also Supplementary Figure 7). Therefore, in this study, we refrain from making inferences based on small changes in allele proportions among populations, but rather restrict this to scenarios where estimated proportion changes from zero to one or from 0.5 ± 0.2 to one.

Notwithstanding the problems of precisely quantifying allele frequency, it is straightforward to identify with confidence those SNVs that were absent in the ancestral population and at very high abundance in one or more of the evolved populations after 300 generations. A variant $m$ was considered absent in the ancestral t0 population if $D_{t0m} >= 10$ and $A_{t0m} = 0$. Since this ancestral population was clonal, the possible allele frequencies that $D_{t0m}$ can take are zero or unity or, if heterozygous, 0.5. Assuming a binomial distribution, the probability of an allele with population frequency of 0.5 being not represented among a sample of 10 aligned sequence reads is $0.5^{10} = 0.00098$. Similarly, if in a t300 population $p$, $D_{pm} >= 5$ and $A_{pm}/D_{pm} = 1$, then it can be confidently inferred that variant $m$ is indeed present in population $p$. Assuming a high sequencing error rate of 1%, the probability of falsely calling the presence of a variant on this basis is $0.01^5 = 1e^{-10}$; in practice, Illumina error rates are closer to 0.1% than 1% making the probability of a false positive even smaller. Thus, we include in our analyses any candidate variant $m$ in a population $p$ if all the following criteria are satisfied:

i.　$D_{t0m} >= 10$
ii.　$A_{t0m} = 0$
iii.　$D_{pm} >= 5$
iv.　$A_{pm}/D_{pm} = 1$
v.　Variant $p$ is called by the bcftools and passes filters in at least one population

The matrices containing SNVs that had passed reliability filtering, had become fixed after 300 generations, and were within protein-coding regions, were used for the downstream statistical analyses implemented in R using the packages vegan (2.4–3), mixOmics (6.1–3), ade4 (1.7–10), topGO (2.26–0, with Bioconductor 3.4), biomaRt (2.3–0 with Bioconductor 3.4), and phangorn (2.2–0).

The matrices were first used for PCAs based on Euclidean distance matrices. From these, we pulled the top 100 loadings for the two major axes of variation, and associated them to a treatment through the plotLoadings functions within the mixOmics package (6.1–3), which performs discriminant PCA and then represent the loading weight of each selected variable on each component, i.e., the highest absolute value indicating the strongest association of a gene with SNVs to a treatment group. This allows us to find treatment associated groups in which the changes in a gene or SNVs are maximal[57,58]. From this, we can infer a number of genes, were SNVs are most likely associated with a treatment. For these, we performed an enrichment analysis for GO using the R package topGO (2.26–0,with Bioconductor 3.4) and biomaRt (2.3–0 with Bioconductor 3.4). We created a "gene universe" as a reference data set, using the entire *T. pseudonana* genome, and matching all gene locus tags with the appropriate GO term for biological processes through protists.ensemble.org. Our test data frame contained the genes that had been drawn from the top 100 loadings of the two major principle component axes, the associated GO terms, and the PCA loadings. Based on these we performed a Kolmogorov–Smirnov-like test which computes enrichment taking into account not only the GO hierarchies but also the PCA loading scores. This allow us to test the over-representation of GO terms within the group of genes found to be most important in the PCA (Table 2).

To test for separation of samples by treatment, and within treatment variation, the Euclidean distance matrices were used to perform Analyses of Molecular Variation (AMOVAs) following ref.[59]. Briefly, the Euclidean distance matrix of SNVs is used to apportion molecular variance according to hierarchical subdivision. In our case, this is variation within populations (biological replicates), between populations (biological replicates within the same treatments), and between treatments. To disentangle whether (i) there were differences in the amount of variation between populations in different treatments and (ii) treatments were significantly genetically divergent from each other, AMOVAs were performed on populations from each treatment separately (yielding a measure of variation that can be compared across treatments) and for all populations in all treatments combined (yielding a measure of variation, or distance, between populations from specific treatments). The significance of the different components of variance was tested using a permutation approach[59], which removes the assumption of normality that underlies conventional ANOVAS, but which is often not appropriate for molecular data. Additionally, to provide explicit significance measures on pairwise differences between treatments, and between treatments and

the ancestor, Permutational Multivariate Analysis of Variance (PERMANOVA) was performed on the same Euclidian distance matrix, where contrasts between treatments were examined through a test of homogeneity of dispersion (permdist in R), followed by TukeyHSD post-hoc tests.

All trees for were built within the phangorn package through neighbor-joining, and annotation of trees was carried out in the FigTree software (FIGTREE 1.4.3 http://tree.bio.ed.ac.uk/software/figtree/).

**Assessment of the abundance of other taxa.** The diatom cultures used in this study were not axenic and therefore the genomic sequence data contained both sequence reads originating from the diatom and also from other organisms such as bacteria that were present in the cultures. To estimate the relative abundances of taxa represented in the data, we used BLASTN (version 2.5.0+) to align 10,000 sequence reads from each sample against the NCBI's non-redundant Nucleotide (also known as "NT") database[60] downloaded on December 9, 2016, using command-line options "-num_alignments 20 -evalue 1e-10". Matches were assigned to taxa using MEGAN (version 5.11.3)[61]. Similar to analysis of the diatom data, we combined the allele frequencies into a single matrix, which was used for the downstream statistical analyses implemented in R using the "vegan" (2.4–3) and "ecodist" (2.01) packages. Distance matrices using the Bray–Curtis index were created from these, on which we ran PERMANOVAs to test for separation of samples by treatment. Pairwise contrasts between treatments were examined in PERMDISP followed by TukeyHSD post-hoc tests. See Supplementary Figure 14 and Supplementary Table 17 for a comparison of bacterial composition across treatments.

**Statistical analyses of fitness trajectories.** The resultant time series of specific growth rates were analyzed using a generalized additive mixed effects model (GAMM) to assess whether the fitness trajectories differed between the selection regimes. We used GAMMs to account for the hierarchical nature of our experimental data. For example, our experimental design yielded replicate fitness trajectories in each selection treatment. This hierarchical structure meant that measurements were non-independent—e.g., measurements from the same replicate will be auto correlated. We account for this by treating replicate as a random effect on the intercept of the model, which models deviations among replicates from the fixed effects as normally distributed with a mean of zero. The most complex models included an effect of treatment (e.g., "22", "26", "32", "22–32") on the intercept (which characterizes the median value of the response variable) and also allowed the shape of the time series, which was modeled using a cubic regression spline, to vary among treatments. Treatment effects on the shape and intercept of the seasonal phenology were modeled as fixed effects in the GAMMs. Model selection entailed fitting a range of models to the data, starting with the full model and then a series of reduced models with interaction terms and main effects removed to test hypotheses about the potential differences in the fitness trajectories among treatments. For multi-model selection, we computed small sample-size corrected AIC scores (AICc) and then compared between models by calculating delta AICc values and AICc weights using the "MuMIn" package. GAMMs were fitted to the data using the "gamm4" package and were conducted in R (v.3.23) (see Supplementary Figure 15 for trajectories per biological replicate).

**Statistical analyses of thermal response curves.** The thermal responses for growth, photosynthesis, and respiration were quantified using a modified version of the Sharpe–Schoolfield equation (see refs.[62,63] for the original equations), which assumes that the rate of growth or metabolism is limited by single enzyme catalyzed reaction

$$\ln(b(T)) = E_a\left(\frac{1}{kT_c} - \frac{1}{kT}\right) + \ln(b(T_c)) - \ln\left(1 + e^{E_h\left(\frac{1}{kT_h} - \frac{1}{kT}\right)}\right) \quad (6)$$

where $b(T)$ is the rate of metabolism (in µg C µg $C^{-1}$ $day^{-1}$) or growth ($day^{-1}$), $k$ is Boltzmann's constant ($8.62 \times 10^{-5}$ eV $K^{-1}$), $E_a$ is the activation energy (in eV) for the metabolic process, indicative of the steepness of the slope leading to a thermal optimum, $T$ is temperature in Kelvin (K), $E_h$ characterizes temperature-induced inactivation of enzyme kinetics above $T_h$ where half the enzymes are rendered non-functional and $b(T_c)$ is the rate of metabolism at an arbitrary reference temperature, here $T_c = 18$ °C, where no low or high temperature inactivation is experienced. Equation (6) yields a maximum metabolic rate at an optimum temperature, where metabolic rates are fastest:

$$T_{opt} = \frac{E_h T_h}{E_h + kT_h \ln\left(\frac{E_h}{E_a} - 1\right)} \quad (7)$$

Equation (6) differs from the Sharpe–Schoolfield equations in several ways. First, we exclude parameters from Eq. (6) used to characterize low-temperature inactivation due to insufficient data to quantify this phenomenon in our analysis. Second, rather than characterize temperature effects below $T_{opt}$ using the Eyring (1935) relation[64], $\left(\frac{T}{T_c}\right)e^{E_a\left(\frac{1}{kT_c} - \frac{1}{kT}\right)}$, we instead use the simpler Boltzmann factor, $e^{E_a\left(\frac{1}{kT_c} - \frac{1}{kT}\right)}$. This simplification enables an explicit solution for $T_{opt}$ (Eq. 7) and

facilitates more direct comparison with previous work on the temperature dependence of metabolism using metabolic theory[65–68].

The parameters $b(T_c)$, $E_a$, $E_h$, $T_h$, and $T_{opt}$, represent traits that characterize a unimodal thermal response curve, and we expect them to differ between selection regimes owing to thermal adaptation. To test this hypothesis, we fitted the rate data to Eq. (6) using nonlinear mixed effects models in the "nlme" package in R. We analyzed the photosynthesis, respiration, and growth data in separate mixed effects models. We also analyzed the thermal responses for ancestor separately from the data for the evolved lineages following the selection experiment. Models included random effects on each of the parameters of Eq. (6) by replicate, and for the analysis with evolved lineages, "selection environment" as a fixed four level factor on each parameter. For the analysis of the metabolism data for the ancestor we included both photosynthesis and respiration data together, with "flux" as a two-level fixed factor on each of the parameters in Eq. (6), to establish whether the thermal responses differed between these fluxes and likely physiological constraints on growth prior to the selection experiment. Model fitting and selection started with the most complex possible model, including fixed and random effects on all parameters. Model selection then proceeded by first removing treatment effects on the parameters individually, then in pairs of two and in pairs of three, and finally, by removing the treatment—effect all four parameters. For multi-model selection we computed small sample-size corrected AIC scores (AICc) and then compared between models by calculating delta AICc values and AIC weights using the "MuMIn" package. When candidate models deviated from the most parsimonious model (that with the lowest AICc score) by less than two AICc units, parameters were averaged across those candidate models. The relative importance of the fixed factors in the averaged model was determined using the sum of their relative weights (see Supplementary Tables 5 and 6).

**Statistical analyses of carbon-use efficiency.** CUE, as the potential for carbon allocation to growth, was calculated from the gross photosynthesis ($P$) and respiration ($R$) data as CUE $= 1 - R/P$. For statistical analysis, we used CUE calculated at the temperature of the selection environment. In the fluctuating treatment, we used data for photosynthesis and respiration at 32 °C to aid comparison with the populations experiencing 32 °C throughout. We fitted a linear mixed effects model to these data, with "selection regime" as a fixed effect and biological replicate as a random effect. Model selection proceeded as described above. See also Supplementary Figure 5.

**Macromolecular composition and photosynthetic efficiency.** The C, N, and P content, the C:N, C:P, N:P, Chl:C ratio, size and silicate content, as well as $\Phi_{PSII}$ at high and low light, were each analyzed using a separate mixed effects model, where "selection regime" was a fixed effect and replicate was a random effect on the intercept (see Supplementary Tables 10 and 11 for details). To quantify the effects of temperature on the elemental composition of the ancestor, a separate mixed effects model was run for ancestral samples only, with "assay temperature" as a fixed factor and replicate as a random effect (see also Supplementary Table 11 for details).

For both datasets, model selection was as described above, with the best model being chosen based on the lowest AICc and high AICc weight.

We quantified the light response curve of $\Phi_{PSII}$ using an exponential decay model

$$\Phi_{PSII} = a * \exp(I * b) \quad (8)$$

where $I$ is the irradiance, $a$ is a normalization constant, and $b$ is the rate constant which characterizes how rapidly $\Phi_{PSII}$ declines with increasing $I$. Eq. (8) was fitted to the $\Phi_{PSII}$ data using a no-linear mixed model including random effects by biological replicate on each parameter and "selection environment" as a fixed factor. Model output was then used to calculate $\Phi_{PSII}$ at the light intensity that samples would have experienced in the incubators (i.e., ~ 100 µmol quanta $m^{-2}$ s$^{-1}$). Model selection then proceeded as described above for $P$ and $R$ data. Detailed model output and model selection data are available in Supplementary Table 9 where nomenclature and acronyms are as given here (see also Supplementary Figure 3).

On the phenotype data, we ran PERMANOVAs to test for separation of samples by treatment. Here, the test is based on the distance matrix containing all phenotypic data of the evolved lineages from all treatments relative to the ancestor at that same temperature. Subsequently, contrasts between treatments were examined through a test of homogeneity of dispersion (permdist in R), followed by TukeyHSD post-hoc tests.

**Data availability.** All sequence data have been deposited in the Sequence Read Archive with accession number SRP114919 (https://www.ncbi.nlm.nih.gov/Traces/study/?acc=SRP114919) and are accessible via BioProject PRJNA397360. All data used for analysis and graphical presentation are available from the corresponding author on request and are uploaded to PANGAEA.

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

## Acknowledgements

This study was funded by a Leverhulme Trust research grant (RPG-2013-335). Whole genome re-sequencing was carried out at Exeter Sequencing Service and Computational core facilities at the University of Exeter, where Dr. Karen Moore, Dr. Audrey Farbos, Paul O'Neill, and Dr. Konrad Paszkiewicz lead the handling of the samples. Exeter Sequencing Services are supported by Medical Research Council Clinical Infrastructure award (MR/M008924/1), Wellcome Trust Institutional Strategic Support Fund (WT097835MF), Wellcome Trust Multi User Equipment Award (WT101650MA), and BBSRC LOLA award (BB/K003240/1). The authors thank Katja Räsänen for helpful suggestions on the role of countergradient variation.

## Author contributions

G.Y.D. conceived and designed the study, analyzed data, and wrote the manuscript. E.S. designed and carried out the experimental work, analyzed data, and wrote the manuscript. A.B. conceived and designed the study. D.S. carried out the bioinformatic analyses of the genomic data. N.S. contributed to the interpretation of the data analysis. All authors contributed to writing the manuscript.
