## [Peer Review File · Nature Communications]

Reviewers' comments:

Reviewer #1 (Remarks to the Author):

This paper examines the temperature adaptation of a model oceanic diatom, *Thalassiosira pseudonana*. It reports on experiments to test the ability of *Thalassiosira* to evolve in response to a range of warming scenarios in order to better understand how global change is likely to impact on the growth and productivity of phytoplankton.

The work is novel, combining both physiological measurements and transcriptome analysis to identify the genes most associated with the changes. Intriguingly, cultures subjected to a fluctuating temperature regime evolved faster than those treated with a steady temperature increment. At the highest temperature tested (32 °C), the growth responses were variable, with an initial increase, a decline and then a rise to control levels after prolonged exposure. Such fluctuations with generation number are reminiscent of changes in response to exposure to elevated CO₂ and ocean acidification (e.g. Li et al *Global Change Biology* 23: 127-137) which imply that changes over the long term give very different results to experiments conducted over a shorter time scale, even if those are in themselves much longer than the acclimation times commonly used in experiments on acclimation to climate change, which may be run for 10 generations or so.

The work is well designed and reported and discussed clearly and thoroughly. My only criticisms are minor:

Phosphorus is mis-spelled throughout the manuscript.

I had problems coming to grips with Figure 2: Why were the "mass-specific rates of P and R normalized to a reference temperature T_c"? If this was done, surely the values obtained would be a ratio and not have the units given on the y axes of the panels in Fig 2?

Reviewer #2 (Remarks to the Author):

The manuscript "Environmental fluctuations accelerate molecular evolution of thermal tolerance in a marine diatom" by Schaum et al. describes the evolutionary response of the model Diatom *Thalassiosira pseudonana* to increases in temperature. In their study, the authors monitored growth of T.p. under 4 different temperature conditions over a duration of 300 generations. An interesting and new approach was the use of fluctuating temperatures changing back and forth between the control temperature and the extreme temperature. After 300 generations, cellular composition, photophysiology, photosynthesis and respiration and genomic analysis via sequencing was performed. Main finding of this study was that under fluctuating temperature, evolutionary rescue was similar fast compared to moderate temperature increases, while cells needed a prolonged time to adapt to extreme conditions.

This study is interesting and important since we urgently need information on how phytoplankton will adapt to climate change. The usual approach of evolution experiments is to either grow cells under the target conditions for a prolonged time or to slowly acclimate them over a certain time. This study does predict that cells acclimated/adapted to a dynamic range could actually adapt faster.

I especially praise that the authors not only measure temperature adaptations by following growth and elemental composition but also try to understand the processes using metabolic traits of PS/Resp and genomic analysis. This process understanding allows to make projections of potentially similar

responses in other phytoplankton species by identifying the importance of well constrained pathways in the temperature response.

While I feel that the study shows very important results and will be a breakthrough for ecological and physiological process studies on evolutionary adaptation, I feel that there are some flaws in the setup and the data presentation and interpretation. I also find that some of the data presentation is confusing or misleading. I would like the authors to rework this manuscript and make the content understandable for a general readership.

In general I think that the data provided in this manuscript are worth being published in a high profile journal, yet, not in the way this MS is written. The conclusion of the data as they are presented is robust (as far as I can judge it), the methods used appropriate and the statistics (I'm not a statistics driven scientist) appropriate.

MAJOR COMMENTS:

My first major comment is regarding the choice of the culture strain and the applied temperature range: the *Thalassiosira pseudonana* (CCMP 1335) strain is maintained by NCMA at 14C and according to the webpage grows between 4 and 25C. Obviously, the authors grew them successfully at higher temperatures and the cultures grew sufficiently fast to conduct this experiment. The authors failed to sufficiently reference their data to previously published data on T.p. temperature effects, e.g. how are their growth rates different to previously published data on temperature effects. Additionally, how would the authors expect the response to be, if cultures e.g. would have been adapted to temperatures between 8 and 25C? It is obvious that in this study the optimum growth temperature is somewhere between 20-30C, yet I do not see information on studies either confirming this or showing otherwise.

Directly related to this: if 300 generations at high or even moderate temp affect the genome, how does a culture isolated in 1958 (according to the culture collection) might have changed?

Both of these comments/questions do not downgrade the importance of the findings of this study and my suggestions might not affect the outcome, but these are concerns which when designing a study have to be considered and should be discussed.

Comment on the experimental design for the fluctuation – In line 44, the authors indicate that temperature variations are an intrinsic feature of natural environments. I do agree (in general) with this statement. Diurnal variations (mainly in coastal regions, estuaries and lakes), seasonal variations and local differences (advection of seawater to higher latitudes) in sea surface temperature are indeed an intrinsic feature, leading to species diversities in the ocean. However, I do not agree with this explanation that the fluctuating temperature studied in this work is mimicking the environment. A temp. change between 22 and 32C every 3-4 generations is hardly happening and thus is an enforced situation with no link to nature and should be defined as such.

I'm not an expert in molecular evolution, as shown in Fig 4a and mentioned in line 172 – it seems that the "evolved" cells had the same trajectory away from the ancestor. Interestingly, both moderate and extreme temperatures are clustered closely together while the moderate temperature cluster is most loosely defined – which I would have expected from the fluctuation adaptation. Can the authors comment on this thought. I might be completely off here, though.

Regarding the basic fluorescence analysis, it seems to me that even after 300 generations, the photosynthetic yield in the dark never recovered to ancestor yields. This general reduction in "fitness" should be discussed.

FURTHER COMMENTS AND QUESTIONS:

Fig. 1A: The fast “adaptation” in the 32C culture after ~200 generations is interesting and should be discussed.

Table 1/Fig 3:

- The data given in Fig. 3 shows the mol:mol based estimate for the cellular compositions. These are based on the concentration of C, N or P per cell basis. In Table 1, volumes (size) of the cells show large differences under high and extreme temperatures. How would the data look like if normalized per volume and not per cell? Please discuss potential consequences on this.
- Especially Photosynthesis and respiration rates (given in per ug C) as well as the corresponding carbon use efficiencies should be reconsidered using a per cell-volume basis.
- I feel that a linear scale rather than a log scale would be more appropriate for the display of these data.
- I feel there is a discrepancy between the data displayed in Fig. 2c vs. the data shown in Table 1. E.g. rates of P , R and CUE (ancestor). Please check these data again.

Line 50: I’m curious how this upper limit for the starting clone was determined

Line 63: I’m puzzled by the continuous increase in growth rate in the control. It is not clear is anything in the acclimation might have changed.

Since the cultures were not axenic what effect could bacteria have, e.g. on metal availability, nutrient availability at the different temperatures?

What is the optimal growth temperature – would light intensity affect this optimum?

Explain the term “evolutionary rescue” early on since this article is supposed to be read by non-specialists.

Line 194: it seems puzzling to me that the high temperature cells are still severely temperature stressed – as also shown in the photosynthetic yield.

How are the average population sizes determined? I assume it is cell density before each dilution averaged over 300 generations but it was not clear from the description.

Line 407: how were the pilot studies conducted. Did the authors pre-acclimate the cultures or have the cultures been transferred to the different temperature immediately?

MINOR COMMENTS (most of which could have been avoided if all authors would have read the MS another time – I likely missed a lot typos, too):

Line 44: add an “in” before adaptation

Line 67: delete comma before ref. 14

Line 505 –In line 435 the authors mentioned Chl determination via acetone extraction, yet in line 505 it was described as methanol extraction. Please correct and clarify

Line 510: delete one on the “was determined”

Line 418 – please indicate which light and how homogeneous the light was distributed. E.g. fluorescent light bulbs are knowingly less bright at the ends and thus can affect cultures.

How were the different temperatures generated? Were the cultures sitting on the lab bench or in a culture room or small incubator? In general, please describe the culture conditions in more detail.

Reviewer #3 (Remarks to the Author):

The manuscript by Schaum et al. describes a potential mechanism by which the diatom *Thalassiosira pseudonana* may quickly evolve enhanced thermal tolerance as the oceans warm. These authors explore the possible role of evolutionary rescue, in which the rise of an adaptive genotype or phenotype is directly associated with recovery to a positive growth after a strong selection. This is an exciting concept that may help explain future microbial growth dynamics in the changing oceans. As outlined below, however, I have three significant areas of concern that the authors should address.

1) Growth rates: The authors monitored growth rates of 6 replicate cultures per temperature treatment for 300 generations, a remarkably labor-intensive undertaking. Perhaps as a compromise, the authors determined growth rates based on flow cytometry counts of cell abundance only twice during each growth cycle "at the beginning and at the end of each transfer." Given that growth rates are determined from the slope of the increase in cell abundance over time, determining growth rates based on only two points will introduce increased measurement error. This raises a second issue. An underlying premise of this work is that beneficial mutations will arise in the cultures (at a rate correlated with cell abundance) and will then become fixed in a population under selective conditions. This implies that there should be a stochastic component to the rise of these mutations. Although the authors have 6 replicates per treatment, they combine the replicate data and report only means. Perhaps they were unable to show results from the individual replicates because of the measurement error? It would have been more convincing to display the data from individual replicates (at least in the supplement) to identify any differences in the timing of increases in growth rates and thus the impact of mutations on rescue. The authors should also discuss potential explanations for the increase in growth rates during the initial weeks of growth at 32°. And finally, the authors state "Co-occurring bacteria, e.g. *Rhodobacter* spp and *Marinobacter* spp are known to be crucial for nutrient recycling and signaling molecule metabolism in *T. pseudonana*. We therefore made no attempt to make the cultures axenic as this would have incurred a substantial fitness cost." The underlying logic behind this decision should be further clarified given that any accompanying bacteria would potentially also be subjected to strong selective pressures under the different temperature regimes. Would the loss of a particular type of bacteria or a change in ratio of the different bacteria under different temperature regimes have an impact on the subsequent growth rate of the diatom? The authors should address whether there were any changes in the bacterial compositions under the different treatments and how this may have impacted the diatom phenotypes.

2) Cell abundance: Figure S2 does not make sense. If I am interpreting the Y-axis correctly, then this data implies that some cultures reached cell abundances as high as 10^{16} cells per ml as in 2SA for the 26° cultures. This is not possible. Perhaps the y axis does not actually depict cells per ml and is instead the number of cells in the flask (volume of flask X cells per ml)? Also, the shape of the different temperature curves is identical to the shape of the growth rate curves of Fig 1. This implies that each culture was transferred on the same day after initiation (e.g., every 3 days) and the cell number plotted is the number at the time of transfer. The authors should discuss the relation between cell abundance in the flask and number of generations of cell divisions.

3) SNP analysis: The authors resequenced the ancestor strain and each of the six replicates from the

different temperature treatments. They aligned the reads from the resequenced strains to the version 2 of the original genome sequence. They then defined as a single allele as those positions within the ancestor strain in which all reads were identical – in other words no SNPs were detected in the sequences for these positions. These “single allele” positions were then examined in the selected strains. The authors should provide more details on how they determined allele frequencies in the subsequent analyses. I believe they counted the proportions of different nucleotides at a given position and defined this as allele frequency at that position. The authors should provide more information about their techniques as the measure I describe is not a true allele frequency and rather is a function of read depth and stochastic sequencing of different DNA molecules within a given sequence mix. There is no discussion of whether any positions were detected where all reads differed from the ancestor. There also is no discussion of how sequence errors were incorporated. A key premise of the concept of evolutionary rescue is that an adaptive allele should become fixed within the population. The sequence data should be discussed within this context. With their methods, how would they define a fixed allele? Finally, although the authors detect SNPs in the resequenced strains, they provide no data that these genes and the different alleles are expressed in the selected strains under the growth conditions. RNAseq data of the selected strains would help in this regard.

Dear editors and referees,

We want to thank the reviewers for their detailed and insightful comments. We appreciate their thoughtful suggestions and have taken great care to address their comments in detail below. Within the manuscript, changes in response to the reviewers' suggestions are marked in blue.

Further, we have adhered to the details given by the editorial board and can clarify that

- Data used for figures and analysis will be available as csv files as part of the supporting information upon acceptance
- Accession numbers for WGS data have been provided (SRA: SRP114919 and BioProject: PRJNA397360) and data will be made publicly available without restriction on acceptance.
- All figures comply with the standard requirements
- We attach the reporting check list for life science articles as a related manuscript file

Yours sincerely on behalf of all authors

Elisa Schaum

Comments from reviewers

Reviewer #1 (Remarks to the Author):

My only criticisms are minor:

Phosphorus is mis-spelled throughout the manuscript.

We have corrected the typographical error throughout the manuscript.

I had problems coming to grips with Figure 2: Why were the "mass-specific rates of P and R normalized to a reference temperature T_c "? If this was done, surely the values obtained would be a ratio and not have the units given on the y axes of the panels in Fig 2?

In Fig. 2 C & D, $P(T_c)$ and $R(T_c)$ are parameters that come from fitting equation 6 to the thermal response curves for photosynthesis and respiration. These quantities are effectively the intercepts (normalisation constants) of the reaction norms – e.g. the rates at 18°C. These values give an indication of the metabolic capacity and provide a means for comparing metabolic rates among the treatments. Note that they are derived directly from the fit of equation 6 and are not calculated from a quotient. We have changed the wording in the figure legend and elsewhere in the text, removing the word 'normalised' to avoid any potential confusion – see lines 459 – 460, 134-135, and 804-805.

Reviewer #2 (Remarks to the Author):

My first major comment is regarding the choice of the culture strain and the applied temperature range: the *Thalassiosira pseudonana* (CCMP 1335) strain is maintained by NCMA at 14C and according to the webpage grows between 4 and 25C. Obviously, the authors grew them successfully at higher temperatures and the cultures grew sufficiently fast to conduct this experiment. The authors failed to sufficiently reference their data to previously published data on T.p. temperature effects, e.g. how are their growth rates different to previously published data on temperature effects. Additionally, how would the authors expect the response to be, if cultures e.g. would have been adapted to temperatures between 8 and 25C? It is obvious that in this study the optimum growth temperature is somewhere between 20-30C, yet I do not see information on studies either

confirming this or showing otherwise.

Directly related to this: if 300 generations at high or even moderate temp affect the genome, how does a culture isolated in 1958 (according to the culture collection) might have changed?

Both of these comments/questions do not downgrade the importance of the findings of this study and my suggestions might not affect the outcome, but these are concerns which when designing a study have to be considered and should be discussed.

We thank the referee for their positive comments and for raising this important issue. Our strain is a clone of the original CCMP strain 1335, and it was sourced not from NCMA but from CCAP, where the culture conditions are between 18°C and 22°C. We have made this clearer in the methods now (see lines 494-496).

Previous experiments with *T. pseudonana* in other laboratories also show that the lineage grows well at temperatures between 16°C and 31°C, with optimum temperatures between 26 to 28°C (e.g. Boyd *et al.* 2013 - doi:10.1371/journal.pone.0063091, Thomas *et al.* 2017 - doi:10.1111/gcb.13641). Furthermore, in these studies optimal growth rates range between 1 and 1.3 divisions per day. These data are consistent with ours. As we show in Fig. 1B, we characterised the thermal tolerance curve of the ancestor (grey points and line), which had an optimum temperature of 28°C and an optimal growth rate of 1 division per day. We have added references to these previous studies on the effects of temperature on *T. pseudonana* in the methods (lines 498, 500-502).

We also agree with the referee that this strain, which has been in long-term culture, may have changed significantly relative to wild-type strains. However, our experiment is a first-step, proof-of-principal study to identify some of the factors that drive the evolution of thermal tolerance in a marine diatom. The study design therefore is a balance between realism and practicality. One of our primary goals was to quantify patterns of molecular evolution using an evolve and re-sequence approach. Consequently, we opted for the strain of *T. pseudonana* that had already had its genome sequenced and annotated to facilitate this approach. We appreciate the referee's point about potential divergence from the wild type and have included reference to this issue in the discussion (see lines 234-242). However, we note that many of the phenotypic and genomic changes we observe in the evolved lineages have occurred in well-conserved metabolic pathways opening up the potential to “to make projections of potentially similar responses in other phytoplankton” (as pointed out by the referee). Repeating this kind of proof-of-principal experiment with freshly isolated strains among diverse taxa will be an important next step in the field.

Comment on the experimental design for the fluctuation; In line 44, the authors indicate that temperature variations are an intrinsic feature of natural environments. I do agree (in general) with this statement. Diurnal variations (mainly in coastal regions, estuaries and lakes), seasonal variations and local differences (advection of seawater to higher latitudes) in sea surface temperature are indeed an intrinsic feature, leading to species diversities in the ocean. However, I do not agree with this explanation that the fluctuating temperature studied in this work is mimicking the environment. A temp. change between 22 and 32C every 3-4 generations is hardly happening and thus is an enforced situation with no link to nature and should be defined as such.

We agree with the referee that the design of our fluctuating treatment does not mirror nature. It was never our intention to precisely mimic natural temperature variation in an experimental setting. Rather our aim was to compare evolutionary responses under stable versus fluctuating conditions, noting that this is an important comparison because natural environments exhibit fluctuations in abiotic conditions, which are likely to be important in mediating adaptive responses to extreme

conditions. We have changed the wording here to make it clear that the design of our fluctuating treatment was not to mirror the natural environment but rather to compare stable versus fluctuating conditions.

I'm not an expert in molecular evolution, as shown in Fig 4a and mentioned in line 172; it seems that the "evolved" cells had the same trajectory away from the ancestor. Interestingly, both moderate and extreme temperatures are clustered closely together while the moderate temperature cluster is most loosely defined; which I would have expected from the fluctuation adaptation. Can the authors comment on this thought. I might be completely off here, though.

The reviewer raises a very interesting point here.

In the molecular data, the replicate selection lines in the moderate warming treatment are more variable than those in the fluctuating treatment. One could also hypothesise that the reverse pattern might be expected (as the referee appears to have done) given that the fluctuating environment is more complex which might be expected to yield greater diversity. The more conserved response in the fluctuating environment might arise from stronger directional selection owing to the regular exposure to extreme conditions, which is likely to impose a stronger selection pressure than the moderate warming treatment. Indeed this explanation is consistent with the trade-off observed in the thermal tolerance curve of the fluctuating treatments (Fig. 2B), where we observe that the evolution of high temperature tolerance comes at the expense of performance at low temperatures. We have included this discussion in the section explaining the patterns of molecular evolution (see lines 192 to 202).

Regarding the basic fluorescence analysis, it seems to me that even after 300 generations, the photosynthetic yield in the dark never recovered to ancestor yields. This general reduction in "fitness" should be discussed.

We have added a sentence stating that some physiological traits never fully recovered in the 32°C-evolved samples (lines 165-167).

FURTHER COMMENTS AND QUESTIONS:

Fig. 1A: The fast "adaptation" in the 32C culture after ~200 generations is interesting and should be discussed.

We agree with the referee and have emphasised this.

Table 1/ Fig 3:

- The data given in Fig. 3 shows the mol:mol based estimate for the cellular compositions. These are based on the concentration of C, N or P per cell basis. In Table 1, volumes (size) of the cells show large differences under high and extreme temperatures. How would the data look like if normalized per volume and not per cell? Please discuss potential consequences on this.

The data in Fig. 3 are ratios of the cellular macromolecular constituents – e.g. N:P; C:P ratios. Because these quantities are ratios, they are independent of whether they are calculated on a per cell or per unit volume basis.

- Especially Photosynthesis and respiration rates (given in per $\mu\text{g C}$) as well as the corresponding carbon use efficiencies should be reconsidered using a per cell-volume basis.

The rates of photosynthesis and respiration are given in units of $\mu\text{g C}$ per $\mu\text{g C}$ per day. This biomass normalisation is the standard way metabolic rates are reported in the algal physiology literature (see Geider et al. 1987 /doi/10.1111/j.1469-8137.1987.tb04788.x; Falkowski et al 1985 doi/10.4319/lo.1985.30.2.0311/) because the units of carbon can be cancelled to obtain a value in units of inverse time thus enabling coupling between rates of metabolism and growth. We prefer to retain these units for ease of comparison between metabolic and growth rates as well as in comparing to other studies in the literature. Furthermore, interpretation of the rates per unit volume is more difficult because we don't know whether bigger cells have more (non-metabolic) vacuole. Nevertheless, Below, we provide a figure of the photosynthesis and respiration rates in μgC per unit bio-volume per day. Note that while this inevitably changes the absolute values at the intercepts and increases the magnitude of the standard errors, the relative differences between the warmed treatments and the ancestor and control are conserved.

- I feel that a liner scale rather than a log scale would be more appropriate for the display of these data.

We assume that the referee is referring to the reaction norms for photosynthesis and respiration in Fig 2, though its not 100% clear. These data are presented on a log-scale because we fitted equation 6 to the data on a log-scale. Thus the figures and associated model fits are an accurate depiction of the analysis. Fitting these non-linear models using mixed effects models is reasonably challenging

computationally and we always observe far better and more consistent model convergence when using the log-transformed version of equation 6. Additionally, log-transformation is important for interpreting the data graphically as it enables us to plot data that span a large range of values on the same scale. When plot on a linear scale a few very large values completely obscure the plot. For these reasons we prefer to keep the plot and the statistical analyses as they are.

- I feel there is a discrepancy between the data displayed in Fig. 2c vs. the data shown in Table 1. E.g. rates of P, R and CUE (ancestor). Please check these data again.

We have now double-checked that all rates are congruent between the table and the figure, e.g. that all rates are reported in μgC per μgC per day – differences in the previous version arose from Fig. 2 being displayed as values per hour, whereas values in the table were calculated over a 24-hour period to account for respiration occurring throughout 24 hours, whereas photosynthesis only occurs in the light (12 hours). All values have now been amended to values per day. Many thanks for spotting this inconsistency.

Line 50: I'm curious how this upper limit for the starting clone was determined.

The upper thermal limit was determined by examining growth responses at temperatures ranging from 16°C to 40°C for 2 transfers (~14 days). These data are presented in Fig 1B – grey curve. We found that at temperatures exceeding 32°C, growth rates were too slow to be used in a long-term selection experiment (we calculated that it would take 4 years to reach a sufficient number of generations which exceeded the period that the grant had been allocated for) and at temperatures above 35°C, mortalities were too high to be promising for batch-culture.

Line 63: I'm puzzled by the continuous increase in growth rate in the control. It is not clear is anything in the acclimation might have changed.

The continual but subtle increase in growth rate in the control is likely driven by adaptation to the laboratory culture regime. Laboratory adaptation is a well-known but poorly documented phenomenon in marine and freshwater phytoplankton. In culture collections in particular, lineages are often grown in dense populations that are only seldom transferred into fresh media, i.e. they live under nutrient-deplete conditions. Evolution experiments such as ours use a semi-continuous batch culture method where populations are maintained a low density by periodic dilution to prevent major changes in nutrient concentrations. Consequently, the continual increase in growth rate in the control likely reflects selection for fast growth rate in this semi-continuous resource replete culture regime. This is why we also compare the cultures evolved under warming to cultures evolved at 22°C, and not only to the ancestor, to control for this laboratory adaptation effect.

Since the cultures were not axenic what effect could bacteria have, e.g. on metal availability, nutrient availability at the different temperatures?

We already state (albeit briefly) that the bacteria likely have an important role to play, and that this role might change as a function of temperature. Sadly, we cannot answer this interesting question in more depth with the data that we have currently available. However, in light of the referee's comment and that of referee 3 (see below) we have used our existing WGS data to investigate the biological community associated with the diatom and found that it is dominated by the bacterial species *Marinobacter salarius*, which belongs to a genus previously known to associate with

diatoms. We found no significant differences in the bacterial communities among the selection regimes. (see Fig. S9, and PERMANOVA in Table S16).

What is the optimal growth temperature; would light intensity affect this optimum?

Optimum growth temperature here is defined as the temperature at which growth is fastest in the short-term (see Fig 1B and table S1). Light intensity will almost certainly have an impact on the optimum temperature, indeed a recent paper by Edwards et al 2016 in *Limnology & Oceanography* (doi: 10.1002/lno.10282) documents how light limitation can reduce the optimum temperature for growth rate. However, investigating this interaction in the context of our selection experiment was beyond the scope of this project

Explain the term “evolutionary rescue” early on since this article is supposed to be read by non-specialists.

We now explain the term early on when it is first mentioned in the main text and in the figure legends (Line 99-101).

Line 194: it seems puzzling to me that the high temperature cells are still severely temperature stressed; as also shown in the photosynthetic yield.

Several lines of evidence suggest that the lineages selected under constant severe warming at 32°C were still temperature-stressed and had not fully adapted to those conditions. First, all else being equal, given the exponential relationship between temperature and growth rate, one would expect higher growth rates in the adapted populations at 32C relative to the adapted populations at 26C, simply because of the 6°C temperature difference. We observe the opposite result the 26°C populations have a growth rate of ~1.5 d⁻¹ by 300 generations, while the 32C populations have a growth rate of ~0.6 d⁻¹. Given the growth rate of 1.5 d⁻¹ at 26C and assuming a Q10 of 2 we would expect fully adapted populations at 32C to have a growth rate of ~2.3 d⁻¹. Clearly the 32C populations are a long way off this. Second, many of the phenotypic traits, like photosynthetic yield and the carbon-use efficiency are lower in the 32C populations than they are in the fluctuating treatments, which experience the severe environment in short bursts followed by periods in benign conditions. Third, the molecular data demonstrate quantitatively that the 32C populations have evolved less and are genetically more similar to the ancestor and the control compared to the populations in the moderate and fluctuating warming treatments. These points are the main focus of the paper – i.e. that the fluctuating regime accelerates adaptation to the severe environment, relative to the populations that experience constant, severe conditions. We have tried to make these points as clear as possible throughout the manuscript.

How are the average population sizes determined? I assume it is cell density before each dilution averaged over 300 generations but it was not clear from the description.

Average cell densities are calculated at each transfer per biological replicate from three technical replicates. We have added this to the methods (Lines 516-517).

Line 407: how were the pilot studies conducted. Did the authors pre-acclimate the cultures or have the cultures been transferred to the different temperature immediately?

We apologise for the confusion here. Pilot studies was misleading. These are simply the thermal tolerance curves for the ancestor, measured in exactly the same way as for the evolved lineages at

the end of the experiment. The populations were pre-acclimated to each assay temperature for one transfer prior to determining the growth rate.

MINOR COMMENTS (most of which could have been avoided if all authors would have read the MS another time; I likely missed a lot typos, too):

Line 44: add an “in” before adaptation

Line 67: delete comma before ref. 14

Line 505 ;In line 435 the authors mentioned Chl determination via acetone extraction, yet in line 505 it was described as methanol extraction. Please correct and clarify

Line 510: delete one on the “was determined”

Line 418 13; please indicate which light and how homogeneous the light was distributed. E.g. fluorescent light bulbs are knowingly less bright at the ends and thus can affect cultures.

How were the different temperatures generated? Were the cultures sitting on the lab bench or in a culture room or small incubator? In general, please describe the culture conditions in more detail.

We thank the reviewer for their detailed and insightful comments and have addressed all grammatical and typographical errors. We have added information on the type of incubators that were used and how varying light conditions were dealt with (see lines 511-514) to the methods.

Reviewer #3 (Remarks to the Author):

1) Growth rates: The authors monitored growth rates of 6 replicate cultures per temperature treatment for 300 generations, a remarkably labor-intensive undertaking. Perhaps as a compromise, the authors determined growth rates based on flow cytometry counts of cell abundance only twice during each growth cycle; at the beginning and at the end of each transfer. ; Given that growth rates are determined from the slope of the increase in cell abundance over time, determining growth rates based on only two points will introduce increased measurement error. This raises a second issue. An underlying premise of this work is that beneficial mutations will arise in the cultures (at a rate correlated with cell abundance) and will then become fixed in a population under selective conditions. This implies that there should be a stochastic component to the rise of these mutations. Although the authors have 6 replicates per treatment, they combine the replicate data and report only means. Perhaps they were unable to shown results from the individual replicates because of the measurement error? It would have been more convincing to display the data from individual replicates (at least in the supplement) to identify any differences in the timing of increases in growth rates and thus the impact of mutations on rescue. The authors should also discuss potential explanations for the increase in growth rates during the initial weeks of growth at 32°.

We have added a figure showing the per-replicate growth trajectories to the supporting information (Figure S8). The figure shows that the variation in the trajectories is reasonably small among replicates and that evolutionary rescue, where it does occur, is largely replicable in magnitude and timing within-treatments. Please note that the statistical method we used to analyse these data, a generalised additive mixed effects model, accounts for both the average trend among replicates within a treatment (via the fixed ‘time’ covariate) as well as replicate-level deviations from this average trend via the random effects (where replicate is nested within treatment). In general the variance among replicates within a treatment is low compared to the variance between treatments, which is why we observe significant treatment effects on the trajectories as well as the median growth rate. Hence, favouring the most parsimonious depiction of the key findings, we opted to present the treatment-level trends in the main text. However, to ensure confidence in the robustness of the results we have included the replicate-level fitness trajectories in the supplement.

One possible explanation for the increase in growth rate during the first 3 weeks of the experiment could be that the initial plastic response to high temperatures is to grow faster (as all processes are sped up at higher temperatures). Fast growth at high temperature may however expose cells to accumulative oxidative damage and can then not be maintained in the long-term through plasticity alone – genetic change through evolution is necessary here to either revert cells from a stressed state or allow them to deal with being in a state of photo oxidative stress. There may also be a high costs associated with the large plastic response of the first two-three transfers. These explanations are however highly speculative and we do not have data to support or refute them. Consequently, we prefer not to include them in the manuscript.

And finally, the authors state that Co-occurring bacteria, e.g. Rhodobacter spp and Marinobacter spp are known to be crucial for nutrient recycling and signaling molecule metabolism in T. pseudonana. “We therefore made no attempt to make the cultures axenic as this would have incurred a substantial fitness cost.” The underlying logic behind this decision should be further clarified given that any accompanying bacteria would potentially also be subjected to strong selective pressures under the different temperature regimes.

Would the loss of a particular type of bacteria or a change in ratio of the different bacteria under different temperature regimes have an impact on the subsequent growth rate of the diatom? The authors should address whether there were any changes in the bacterial compositions under the different treatments and how this may have impacted the diatom phenotypes.

As mentioned also in our response to reviewer 2, this is a very interesting question. We know from the flow cytometry data, that total abundance of bacteria was not significantly different throughout time or between treatments.

The whole-genome shotgun resequencing data offers an opportunity to partially address these issues, but with some important caveats. Since the subject of the study was the diatom rather than the whole microbiota, we chose DNA-extraction methodology that would enrich for diatom DNA whilst minimising the ‘contamination’ by bacteria. Nevertheless, DNA sequence results contained large numbers of sequence reads that closely matched known bacterial genome sequences. We made no efforts to avoid variation in efficiency of bacterial DNA capture and there may have been ascertainment bias and thus the representation of each bacterial genome in the sequence data might not correlate closely with the actual abundance of that bacterium in the culture; quantitative inferences of microbial profiles in this experiment must be treated extremely cautiously. However, some consistent patterns were apparent. For example, among the sequence reads that don’t match the diatom genome, BLASTN matches were dominated by hits against just a few bacterial taxa, namely the gammaproteobacterial species *Marinobacter salarius* and genus *Pseudomonas*, the alphaproteobacterial order Rhizobiales and family Rhodobacteraceae, and the order Flavobacteriales. The only bacterial species accounting for > 10% of the sequence reads was *Marinobacter salarius*, a species previously reported as being found with diatoms. We analysed the bacterial community composition data using a PCA and a PERMANOVA and found no discernible pattern or association between bacterial abundance profiles and thermal treatment regimes (see Fig S9 and Table S16).

2) *Cell abundance: Figure S2 does not make sense. If I am interpreting the Y-axis correctly, then this data implies that some cultures reached cell abundances at high as 10 to the 16 cells per ml as in 2SA for the 26° cultures. This is not possible. Perhaps the y axis does not actually depict cells per ml and is instead the number of cells in the flask (volume of flask X cells per ml)? Also, the shape of the different temperature curves is identical to the shape of the growth rate curves of Fig 1. This implies that each culture was transferred on the same day after initiation (e.g., every 3 days) and the cell number plotted is the number at the time of transfer. The authors should discuss the relation between cell abundance in the flask and number of generations*

of cell divisions.

The axes in Fig S2 are using the natural logarithm, not log10 values. We now make this clear. This means that a value of 13.5 (max value in in 26C treatment) is equivalent to $\sim 7 \cdot 10^5$ cells mL⁻¹ which is about what we would expect for a very fast-growing culture.

It is expected that the population growth trajectory would mirror that of the growth rate trajectory when we only display the densities at the end of transfer. When we include the starting density (which is the same across all treatments and time points) for each transfer the information then is contained is the change in the steepness of the slope at each transfer. While valid and used in other papers (see, for example Hao et al 2015 doi: 10.1111/ele.12465), we found the current version easier to interpret visually but we are happy to provide a modified version instead if the referee prefers.

3) SNP analysis: The authors resequenced the ancestor strain and each of the six replicates from the different temperature treatments. They aligned the reads from the resequenced strains to the version 2 of the original genome sequence. They then defined as a single allele as those positions within the ancestor strain in which all reads were identical; in other words no SNPs were detected in the sequences for these positions. These “single allele “ positions were then examined in the selected strains.

Yes, the reviewer’s understanding of our methodology is correct.

The authors should provide more details on how they determined allele frequencies in the subsequent analyses.

We have now added the following text (see lines 712 onward) to more clearly describe our approach: “we counted the proportions of the different nucleotides at a given position and defined the estimated allele frequency as this count.”

To clarify further, please note that for some analyses, we aggregated individual SNP sites together by gene. In this case, we did not calculate an allele frequency for each gene. Rather, we used a metric of genetic divergence from the ancestral genotype. This metric consisted of the sum of the frequencies of all non-ancestral-type single-nucleotide variants within that gene, necessarily bounded by 1.

The authors should provide more information about their techniques as the measure I describe is not a true allele frequency and rather is a function of read depth and stochastic sequencing of different DNA molecules within a given sequence mix

Yes, the reviewer is correct that this is an estimate rather than a precise measurement and we did not previously state this explicitly. We now have added the following text to lines 714 to 715: “This is an estimate rather than a precise measurement because the observed frequency of sequence reads is a function of several factors including sampling error, sequencing bias, sequencing errors as well as the frequencies of the different DNA molecules in the sample.”

I believe they counted the proportions of different nucleotides at a given position and defined this as allele frequency at that position.

Yes, the reviewer is correct in inferring this. We have now made this absolutely explicit in the Methods sections, writing “we counted the proportions of the different nucleotides at a given position and defined the estimated allele frequency as this count.” in line 713.

There also is no discussion of how sequence errors were incorporated.

As the referee points out, inevitably there will be sequencing errors. We have now added some text to the Methods (lines 717 to 726) to highlight this caveat and to justify our approach: “The major types of sequencing errors in this platform are substitution miscalls. Rate of sequencing errors are expected to be low, especially after trimming and filtering, which left almost all of the base-calls in our data with a Phred-like quality score of > 30 (representing an error rate of less than 1 in 1000, consistent with the range reported in previous studies of error rates in Illumina sequence data. Nevertheless, in the analyses presented here, which focus on overall trends and relationships among samples based on estimates of frequencies across large numbers of SNPs, we do not expect sequencing errors to present a significant issue. Neither the random nor the systematic sequencing errors should be associated with any specific treatment. When interpreting results of genotyping any individual locus, however, consideration of technical errors would be of more importance.”

A key premise of the concept of evolutionary rescue is that an adaptive allele should become fixed within the population.

According to Gonzalez et al. (2013) *Philos Trans R Soc Lond B Biol Sci.* 368: 20120404, evolutionary rescue is defined as scenarios where evolution can reverse demographic threats due to environmental stress, and so prevent extinction. This definition does not require fixation, but much of the literature about evolutionary rescue does indeed deal with fixation. However, in principle, a population could be rescued from extirpation by maintenance of heterogeneity (multiple alleles); directional selection is, at least in theory, not the only route by which the population could avoid demise. This could be mediated by mechanisms including disruptive selection, diversifying selection and heterozygous advantage. Furthermore, it is possible that at the t300 time-point, the population is on a trajectory towards fixation but has not yet reached it but is still better adapted to the elevated temperature than its antecedents. Additionally, both directional selection (where we would expect fixation) or diversifying selection (where we may not see fixation) could contribute to adaptation, or the time frame used here may simply not have been long enough for equilibrium conditions to be reached. Therefore, in our analyses we have opted to include all variable sites (SNPs) even where the estimated frequency is lower than 100% (and are therefore, by definition, not fixed). We now highlight which of the 20 genes most strongly associated with each of the principal components (and therefore the various treatments) are ‘fixed’ in at least one of the replicate populations in table S14 and in Figure 4 (panel C).

The sequence data should be discussed within this context. With their methods, how would they define a fixed allele?

As discussed above, we believe that all variants should be considered in our analysis, not just the fixed ones. A reasonable definition of a fixed allele would be “a fixed allele is an allele that is the only variant that exists for that gene in all the population.” We are dealing here with single-nucleotide positions rather than genes. In our SNP-centric analyses, we could operationally define a fixed allele as a single-nucleotide site in the genome where only one variant exists for that site, and we would exclude sites where that only one variant is the same variant also present in the ancestor. A subset of our observed SNPs show 100% consensus among the reads within at least one of the t300 samples; therefore, these could be considered to be fixed in at least one of the samples – we highlight which of the 20 genes most strongly associated with each of the principal components (and therefore the various treatments) are ‘fixed’ in table S14. Overall, Of the 12,823,890 SNPs considered in this analysis, 1970 showed 100 % consensus in aligned reads. That is, 0.015% of the

SNPs appear to have become fixed in at least one of the t300 populations. Of course this estimate is not precise as sequencing errors will cause slight underestimation of fixation and stochasticity in shallow sequence alignments will cause slight overestimation fixation.

There is no discussion of whether any positions were detected where all reads differed from the ancestor.

There were 1970 single-nucleotide sites where, for at least one of the t300 populations, all of the reads were in agreement and different from the ancestor. As discussed above, these could be considered to be 'fixed'. We have added this information to table S14 in the supporting information.

Finally, although the authors detect SNPs in the resequenced strains, they provide no data that these genes and the different alleles are expressed in the selected strains under the growth conditions. RNAseq data of the selected strains would help in this regard.

We agree that RNA sequencing would make an interesting future contribution. However, given that many of the genes under selection have unknown function, including RNAseq at this stage may contribute little additional mechanistic insight. A more functional genomics approach to these questions is an important aspiration for future work, but will take years to complete. We hope the referee agrees that whilst incomplete, the work presented here is an important step in this direction.

Reviewers' comments:

Reviewer #1 (Remarks to the Author):

I have happy with the changes the authors have made in response to my, relatively minor, criticisms. I think this manuscript will make a useful and novel contribution to the literature

Reviewer #2 (Remarks to the Author):

The authors did a good job revising the manuscript. The authors answered all my questions and revised the manuscript according to most of my suggestions/critiques. I have no additional critiques.

Reviewer #4 (Remarks to the Author):

Reviewer 3's comments and concerns on the SNP analysis are quite reasonable while the answers from the authors are not satisfactory.

Precisely, to claim the key statement that environmental fluctuations accelerate molecular evolution of a marine diatom, in place of a rigorous allele frequency of each SNP site, the authors used an "approximate" frequency, which is the average frequency in a population after 300 generations of selection. Reviewer 3 has a serious concern that the approximation could be erroneous, which is reasonable because in Supplementary Table 15, the authors collected only ~18-fold sequence depth of Illumina sequencing reads from a mixture of each population. The authors could have collected the same amount of DNA data from each of many isolated samples in the population to determine SNPs in each sample and then could have determined allele frequencies of SNPs. The genome size of the focal marine diatom, *Thalassiosira pseudonana*, is only ~34 Mbp, and sequencing 100 samples would be feasible. If the authors think it labor intensive to sequence many samples separately, they can collect, say ~100-x sequence coverage data as a substitute to have a more accurate approximation.

In response to Reviewer 3's comments, the following statements were added to the method sections without any rationale behind them. The authors should provide a statistical analysis on why they can make the following statements, or replace an approximate allele frequency with a rigorous one by sequencing individuals in a population.

722-725

"Nevertheless, in the analyses presented here, which focus on overall trends and relationships among samples based on estimates of frequencies across large numbers of SNPs, we do not expect sequencing errors to present a significant issue. Neither the random nor the systematic sequencing errors should be associated with any specific treatment."

The definition of a "fixed" allele is not reliable considering the imprecise definition of allele frequency in this paper.

To confirm the importance of Figure 4C, Reviewer 3 suggests to perform RNA-seq sequencing to see how genes and their different alleles are expressed under the growth conditions. However, the authors reply that many of genes under selection have unknown function, and including RNAseq at this stage may contribute little additional mechanistic insight, which is somewhat confusing.

Response to editor and reviewers:

Dear editorial team,
we are pleased that reviewers 1 and 2 have no further concerns regarding the manuscript in its current state, and that they find it novel and interesting. Reviewer 4 (who replaces reviewer 3) raises an important point with regards to how we had analysed the SNP data. Briefly, they are right in that basing inferences on estimated allele frequencies can entail a large margin of error at lower sequence coverage. We have now made considerable revisions to this analysis and take a much more conservative approach.

To address this, we have re-analysed the data using a more stringent approach of quality control. We apply a statistical technique from Good et al. 2017 Nature doi:10.1038/nature24287, which we use to estimate the proportion of a given SNV in the population. Given the large confidence intervals in some of our samples with low coverage, we now only consider SNVs that have gone to fixation in at least one population of the evolved lineages (as previously suggested by the original reviewer 3) and which we can have a high degree of confidence that variants were either absent or heterozygous in the ancestor and only rose to fixation in the evolved lineage. For these, we now also report the 95% confidence intervals, to show that even in cases where the confidence intervals are wide, the interpretation of our data is not fundamentally changed from the output of the new, more conservative analysis, or from the results that we presented in the previous version of this manuscript.

We provide detailed information on the new analysis below and in the new materials and methods in the main manuscript. We also include information on large-scale changes in copy number and losses of heterozygosity in the new version of the manuscript. We hope that taken together, these will convince reviewer #4 that our analysis of the patterns of molecular evolution are robust.

We have also moved some of the phenotypic data from the supplementary material to the main text to emphasise comparisons between short-term acclimation responses in the ancestor and long-term evolutionary responses in a range of phenotypic traits (see Fig. 3). We have also expanded the discussion of these data (lines 156-199), which we feel significantly increases our understanding of the processes that set the limits of thermal tolerance (*via* acclimation) and determine the evolution of increased heat tolerance in the long-term.

All changes are highlighted in blue. Our point-by-point response to the referee can be found below.

Yours sincerely, on behalf of all authors
Elisa Schaum

Reviewer 4 comments:

Reviewer 3's comments and concerns on the SNP analysis are quite reasonable while the answers from the authors are not satisfactory.

*Precisely, to claim the key statement that environmental fluctuations accelerate molecular evolution of a marine diatom, in place of a rigorous allele frequency of each SNP site, the authors used an "approximate" frequency, which is the average frequency in a population after 300 generations of selection. Reviewer 3 has a serious concern that the approximation could be erroneous, which is reasonable because in Supplementary Table 15, the authors collected only ~18-fold sequence depth of Illumina sequencing reads from a mixture of each population. The authors could have collected the same amount of DNA data from each of many isolated samples in the population to determine SNPs in each sample and then could have determined allele frequencies of SNPs. The genome size of the focal marine diatom, *Thalassiosira pseudonana*, is only ~34 Mbp, and sequencing 100 samples would be feasible. If the authors think it labor intensive to sequence many samples separately, they can collect, say ~100-x sequence coverage data as a substitute to have a more accurate approximation.*

Detailed comments to reviewer 4:

We sincerely thank the referee for their critical assessment of our work. We agree with the reviewer that the previous approach of analysing the estimated SNV frequencies could have led to erroneous interpretation of the results. We have now made extensive reanalyses of our genomic data and take a far more conservative approach by (a) estimating the statistical probabilities of each SNV population proportion and (b) only retaining SNVs that were either absent or heterozygous in the ancestor and subsequently rose to fixation in at least one population of the evolved lineages. It is simply not possible for us to either isolate 100 clones from each replicate of each treatment, which would be $100 \times 6 \times 4 = 2400$ samples to run on the Illumina sequencer! Nor is it possible to redo each population at 100-fold coverage. We simply do not have the resources to leverage such an analysis. Instead we have taken a much more conservative approach to analysing the existing data. This approach doesn't fundamentally change the main interpretation of the results, which gives us confidence in the patterns we are observing.

Below, we provide details on the new analysis, which have also been added to the materials and methods (lines 212-243 and 810-920)

In our analysis, we took two distinct approaches to surveying variation: (i) identifying SNVs that were undetectable in the ancestral population (at a sequencing depth of at least 10 x) but were supported by 100% of the aligned reads in an evolved population (at depth of at least 5 x) and (ii) identifying SNVs that plausibly had a population proportion of 0.5 in the ancestral population (due to heterozygosity), at depth of at least 10 x, and were supported by 100% of the aligned reads in an evolved population, at depth of at least 5 x (see also Table S15 for average sequencing depths).

The output of SNV-calling was a matrix of estimated allele frequencies at each SNV site for each sample. All SNVs included in the matrices had passed reliability filtering, as described

above. Following the approach of a recent study (Good et al. 2017 Nature doi:10.1038/nature24287), the estimated allele frequency was defined as the ratio of sequence reads matching the alternative allele versus total number of reads aligned at that site. In other words, the sample proportion is used as an estimate of the proportion:

$$f_{pm} \approx \frac{A_{pm}}{D_{pm}}$$

where for mutation m and population p , A_{pm} and D_{pm} are respectively the count of aligned reads matching the alternate allele and the total counts (*i.e.* depth). Inevitably, this estimator of f_{pm} will be confounded by sampling error. Assuming that each sequence read is drawn randomly from the population, the sample proportion will follow a binomial distribution:

$$A_{pm} \sim \text{Binomial}(D_{pm}, f_{pm})$$

Given the sample size (in this case, D_{pm}) and the population proportion (f_{pm}), it is possible to calculate the 95% confidence interval of the estimate of f_{pm} (<http://www.jstor.org/stable/2331986>). With small sample sizes, the confidence intervals are wide. For example, with D_{pm} (depth) of 10 x and f_{pm} (allele frequency) of 0.5, the 95% confidence interval is 0.5 ± 0.31 . For depth 50 x it is 0.5 ± 0.14 and for depth 100 x it is 0.5 ± 0.10 (see also Fig. S7). Therefore, in this study, we refrain from making inferences based on small changes in allele proportions among populations, but rather restrict this to scenarios where estimated proportion changes from zero to one or from 0.5 ± 0.2 to one.

Notwithstanding the problems of precisely quantifying allele frequency, it is straightforward to identify with confidence those SNVs that were absent in the ancestral population and at very high abundance in one or more of the evolved populations after 300 generations. A variant m was considered absent in the ancestral t_0 population if $D_{t_0m} \geq 10$ and $A_{t_0m} = 0$. Since this ancestral population was clonal, the possible allele frequencies that D_{t_0m} can take are zero or unity or, if heterozygous, 0.5. Assuming a binomial distribution, the probability of an allele with population frequency of 0.5 being not represented among a sample of 10 aligned sequence reads is $0.5^{10} = 0.00098$. Similarly, if in a t_{300} population p , $D_{pm} \geq 5$ and $A_{pm}/D_{pm} = 1$, then it can be confidently inferred that variant m is indeed present in population p . Assuming a high sequencing error rate of 1%, the probability of falsely calling the presence of a variant on this basis is $0.01^5 = 1e^{-10}$; in practice, Illumina error rates are closer to 0.1% than 1% making the probability of a false positive even smaller. Thus, we include in our analyses any candidate variant m in a population p if all the following criteria are satisfied:

- i. $D_{t_0m} \geq 10$
- ii. $A_{t_0m} = 0$
- iii. $D_{pm} \geq 5$
- iv. $A_{pm}/D_{pm} = 1$
- v. Variant p is called by the *bcftools* and passes filters in at least one population

The matrices containing SNVs that had passed reliability filtering, had become fixed after 300 generations, and were within protein-coding regions, were used for the downstream statistical analyses implemented in R using the packages *vegan* (2.4-3), *mixOmics* (6.1-3), *ade4* (1.7-10), *topGO* (2.26-0, with Bioconductor 3.4) and *biomaRt* (2.3-0 with Bioconductor 3.4), and *phangorn* (2.2-0).

We have now also added an additional analysis to test for enrichment of GO terms relating to biological processes. Using the principal components analysis we identified the 100 SNVs associated with the major axes of variation. We then identified which genes these SNVs correspond to and assessed whether this gene set was significantly enriched in any GO terms. We found that GO terms related to transcriptional regulation, cellular response to oxidative stress and redox homeostasis among others were enriched in the gene set that best characterises the genomic variation among the evolved treatments.

Reviewers' comments:

Reviewer #4 (Remarks to the Author):

With regarding the problem of SNV analysis, because collecting a new substantial dataset is infeasible, the authors newly selected reliable fixed SNVs using a couple of computational methods, which reduced 220k SNVs to 1703 reliable ones. This selection is reasonable considering their limited biological resource. With the revised small set of SNVs, they state that the interpretation of their data is not fundamentally changed from the results in the previous manuscript. Put another way, the old and new versions Figure 4A are fundamentally identical.

However, we can see a significant difference between the two versions. In the old version with unreliable SNVs, the ancestor and the clusters of 26C, 22C, 32C, and fluctuations were clearly separated each other, but they are difficult to distinguish in the new set of reliable SNVs. Moreover, the upper right portion of Supplementary Figure S7 displays that they are hard to be separable even if all fixed variants with upper confidence interval are used. The values in the x-axis and y-axis in the new PCA analysis is much smaller than those in the old one, implying that the separation is subtle according to reliable SNVs. Figure 4C may intuitively suggest a transition from hetero- to homozygosity; however, it is hard to assess the significance without a statistical evidence.

Overall, in the revision, they fail to identify significant genetic divergence among populations adapted to the different warming regimes and the ancestral lineages, which is a key claim given in the abstract. To reinforce their important observations, I would suggest that they should discuss the statistical significance of their observations because they have a limited number of reliable SNVs.

I found no answers to the following question in my previous comments:

"To confirm the importance of Figure 4C, Reviewer 3 suggests to perform RNA-seq sequencing to see how genes and their different alleles are expressed under the growth conditions. However, the authors reply that many of genes under selection have unknown function, and including RNAseq at this stage may contribute little additional mechanistic insight, which is somewhat confusing. "

Minor comments:

* In the legend of Figure 4C, describe that 1703 SNVs are 1689 missense and 14 stop codon variants.

* In the first page of Supplementary Information, use the same title as that in figure legend. For example, the titles of Figure S7 are inconsistent.

Response to editor and reviewers:

Dear editorial team,
we are pleased that our manuscript remains of interest for publication with *Nature Communications*. Below we clarify the points raised by Reviewer 4. We think that the Reviewer may have missed our new Table 1, which is the output of an Analysis of Molecular Variance, and explicitly states that the treatments are significantly different from one another and from the ancestor based on the SNV data. We further address the concerns about the statistical analyses associated with the transitions between homo- and heterozygosity.

All new changes are highlighted in blue. Our point-by-point response to the referee can be found below.

Yours sincerely, on behalf of all authors
Elisa Schaum

Reviewer #4 (Remarks to the Author):

With regarding the problem of SNV analysis, because collecting a new substantial dataset is infeasible, the authors newly selected reliable fixed SNVs using a couple of computational methods, which reduced 220k SNVs to 1703 reliable ones. This selection is reasonable considering their limited biological resource. With the revised small set of SNVs, they state that the interpretation of their data is not fundamentally changed from the results in the previous manuscript. Put another way, the old and new versions Figure 4A are fundamentally identical.

However, we can see a significant difference between the two versions. In the old version with unreliable SNVs, the ancestor and the clusters of 26C, 22C, 32C, and fluctuations were clearly separated each other, but they are difficult to distinguish in the new set of reliable SNVs. Moreover, the upper right portion of Supplementary Figure S7 displays that they are hard to be separable even if all fixed variants with upper confidence interval are used. The values in the x-axis and y-axis in the new PCA analysis is much smaller than those in the old one, implying that the separation is subtle according to reliable SNVs.

We agree with the reviewer that the PCAs with all SNVs and the more conservative approach we now take are not absolutely identical in quantitative terms. This was not the point we were trying to make in the previous rebuttal. The data show that the same qualitative patterns remain using both datasets. For example, the 26 and FS treatments exhibit the most variance and have centroids that are further from the control and the ancestor than the 32°C treatments. These general patterns are conserved between both analyses, which is why we are confident that they are robust.

With respect to the point the referee makes about statistical separation of the treatments, there are several key points. First, the ellipses in PCA give the 95% confidence interval around the centroid and thus give an idea of whether the treatments differ significantly or

not. Second, using the SNV matrices we carried out an Analysis of Molecular Variance to test for significant genetic divergence among treatments as well as among replicates within each treatment. These results were given in Table 1 of the revised manuscript and replaced the Permutational Analysis of Multivariate Variance (PERMANOVA) from the original submission. We have now also included the PERMANOVA in the supplement to support the AMOVA because this allows us to carryout post-hoc pairwise contrasts to identify precisely which treatments differ from one another. The results of the AMOVA are given in Table 1 and those of the PERMANOVA are in Table S12. These analyses identify significant differences among treatments that support the 95% confidence ellipses in the PCA – note that all treatments differ significantly from one another except 26°C and the fluctuating treatment (FS). Again, this confluence in the results gives us confidence that the patterns we are reporting are robust, as we can see that in both the old and the new versions of the analyses the treatments that do and do not overlap are consistent.

With regards to the comment about Fig S7, in line with our reply above, the comparison of the upper and lower 95% confidence intervals around the population proportion of each SNV can only be interpreted qualitatively. Again, the patterns are largely consistent between the estimated population proportion and the upper and lower 95% CI – i.e. the same treatments overlap and/or are distant from one another. Their exact quantitative position in PCA space differ, but that is to be expected because each analysis uses slightly different data. However, it is the broad patterns between treatments that are important and they remain highly conserved between all the analyses.

Figure 4C may intuitively suggest a transition from hetero- to homozygosity; however, it is hard to assess the significance without a statistical evidence.

Out of a total of 7383 SNVs that showed apparent fixation, including 1215 in protein coding genes, only 9 occurred at non-zero frequency in the sequence data from the ancestral population. In other words, the majority (i.e. 7374 / 7383) of these fixed variants were detectable in the ancestral clone.

Therefore, we claim that most of the fixed SNVs originate from the standing variation in the ancestral clone rather than originating from *ab initio* mutation. The only reasonable grounds on which to dispute this statement would be that

- (1) most of our candidate fixed SNVs are false positives or that
- (2) most of the SNVs that we failed to detect in the ancestral clone were false negatives, or that
- (3) the ancestral clone was in fact genetically heterogeneous and contained standing variation at a population frequency too low to be detected by the sequencing.

Below, we explain why these three points seem very unlikely:

- (1) We describe in detail in the methods that we took steps to minimise the rate of falsely calling fixed SNVs. Specifically, in order to be counted as ‘fixed’, the variant had to be present in 100% of the aligned reads and there had to be a read depth of at least 5. It is, of course, possible that an allele frequency of less than 100 % could occur in 100% of the reads. As described in the methods, sequencing from the

population can be reasonably modelled as a binomial process where n =number of reads (i.e. sequencing depth) and probability of success, p =population frequency of the variant. If $p=0.9$ and $n=5$, then the probability of all five reads showing the variant would be $0.9^5 = 0.59$. Therefore, it is indeed true that some of our putatively fixed SNVs might only occur at 90% rather than 100% frequency. If $p=0.5$ and $n=5$, then probability of obtaining five reads with the variant = $0.5^5 = 0.03$, illustrating that alleles further from fixation are less likely to be falsely called as fixed. However, these numbers are a worst-case scenario, in that they are for the minimum sequencing depth of 5x. The boxplots below show the distributions of sequencing depths at the fixed SNVs across each sample (t300_22_b3.depth, t300_22_b6.depth, t300_26_b3.depth, t300_26_b6.depth, t300_32_b3.depth, t300_32_b6.depth, t300_FS_b3.depth, t300_FS_b6.depth, t300_22_b1.depth, t300_22_b4.depth, t300_26_b1.depth, t300_26_b4.depth, t300_32_b1.depth, t300_32_b4.depth, t300_FS_b1.depth, t300_FS_b4.depth, t300_22_b2.depth, t300_22_b5.depth, t300_26_b2.depth, t300_26_b5.depth, t300_32_b2.depth, t300_32_b5.depth, t300_FS_b2.depth, t300_FS_b5.depth).

The summary statistics for sequencing depths across all of the putatively fixed SNV sites were:

Min.	1st Qu.	Median	Mean	3rd Qu.	Max.
0.00	38.00	48.00	49.18	59.00	222.00

So, 75% of the sequence depths under consideration are 38x or above. The probability of all 38 reads showing the variant allele when $p=0.9$ and $n=38$ is $0.9^{38} = 0.018$. Therefore, although we certainly do not claim that none of the putatively fixed SNVs are false positives, the false positives account for a minority of these putatively fixed SNVs.

- (2) Our criterion for calling the variant as absent in the ancestral population was that sequencing read depth was at least 10x in the ancestral sample and that zero of those ≥ 10 reads showed the variant. Assuming that the ancestral population consisted of a single genetically homogeneous clone and that its genome is diploid, then p can take one of only three possible values: $p=0$ (the variant is absent), $p=0.5$ (the variant is heterozygous) or $p=1$ (the variant is homozygous). So, in the case of a false positive, $p=0.5$ or $p=1$. Based on a binomial model with $n=10$, the probability of obtaining zero reads showing the variant given $p=0.5$ is $0.5^{10} = 0.00097$. And this becomes even less probably with sequencing depths of greater than 10x. So, by far

the majority of the SNVs that we failed to detect at $\geq 10x$ are indeed absent from the ancestral population

- (3) We can be positive that the ancestral population was indeed clonal, as great care was taken to dilute the stock population so that only a single cell was propagated to start the experiment. The dilution was confirmed both through flow cytometry and microscopic observation. At the point of DNA extraction from the ancestor, about 10-20 generations would have passed – this is in line with the timeframe in which we would expect to see acclimation, but not evolutionary responses. Extremely rapid evolution and hence meaningful genetic diversification within this time period is not entirely impossible, but given the estimated population sizes and mutation rates under control conditions, remains highly unlikely.

Given the above refutation of the above three arguments, we are confident in our assertion that the majority of variants reaching fixation (or close to fixation) were present in the standing variation of the ancestral clone as heterozygous variants. We are not sure what further statistical test the referee would like to see to demonstrate this any more effectively than we have done already.

Overall, in the revision, they fail to identify significant genetic divergence among populations adapted to the different warming regimes and the ancestral lineages, which is a key claim given in the abstract. To reinforce their important observations, I would suggest that they should discuss the statistical significance of their observations because they have a limited number of reliable SNVs.

We believe that there must have been a slight misunderstanding here. As mentioned above, we have carried out numerous statistical tests (AMOVA, PERMANOVA, PCA) on the SNV data and detect clear, highly significant genetic divergence among treatments. These results are reported in Table 1, Table S12 and Fig. 4A.

*I found no answers to the following question in my previous comments:
“To confirm the importance of Figure 4C, Reviewer 3 suggests to perform RNA-seq sequencing to see how genes and their different alleles are expressed under the growth conditions. However, the authors reply that many of genes under selection have unknown function, and including RNAseq at this stage may contribute little additional mechanistic insight, which is somewhat confusing.”*

We apologise for missing this comment in the last iteration. RNAseq data may indeed yield some additional insight. However, it is simply not feasible for us to carry out such analyses with the resources we have at our disposal. We note that the manuscript already contains a great many different forms of measurement and analysis – e.g. we have run detailed physiological assays, characterised biochemical composition, run evolution and acclimation assays, and re-sequenced the genomes of the ancestor and the evolved lineages. Ultimately there is a limit to the number of measurements that can be made and analysed within the context of a single project and manuscript. We believe that the array of techniques we have applied yields significant and novel insights as commented upon by all 4 referees. However,

there is always more to do and RNAseq would undoubtedly be a good candidate for the next steps.

Minor comments:

** In the legend of Figure 4C, describe that 1703 SNVs are 1689 missense and 14 stop codon variants.*

This information has now been added to the Figure legend of Figure 4C.

** In the first page of Supplementary Information, use the same title as that in figure legend. For example, the titles of Figure S7 are inconsistent.*

We have made sure that all titles are consistent throughout the list of figures and the Supporting Information file.

REVIEWERS' COMMENTS:

Reviewer #4 (Remarks to the Author):

In the response letter, the authors answered all of my questions and comments adequately.

Figure 4a is still difficult to understand. It would be helpful to readers if you explicitly mention, in the figure legend, that the statistical significance of separation of the treatments can be found in Table 1.

The statistical analysis that the majority of fixed variants are detectable in the ancestral clone should be described in the paper.